# scPair: Boosting single cell multimodal analysis by leveraging implicit feature selection and single cell atlases

Hongru Hu [1,2] ✉ & Gerald Quon [2,3] ✉

Multimodal single-cell assays profile multiple sets of features in the same cells and are widely used for identifying and mapping cell states between chromatin and mRNA and linking regulatory elements to target genes. However, the high dimensionality of input features and shallow sequencing depth compared to unimodal assays pose challenges in data analysis. Here we present scPair, a multimodal single-cell data framework that overcomes these challenges by employing an implicit feature selection approach. scPair uses dual encoder-decoder structures trained on paired data to align cell states across modalities and predict features from one modality to another. We demonstrate that scPair outperforms existing methods in accuracy and execution time, and facilitates downstream tasks such as trajectory inference. We further show scPair can augment smaller multimodal datasets with larger unimodal atlases to increase statistical power to identify groups of transcription factors active during different stages of neural differentiation.

Single cell assays have been developed to capture diverse aspects of genome regulation, including gene expression[1,2], chromatin accessibility[3], and methylation profiling[4,5], among others[6,7]. These single modality assays that capture a single data type have been widely deployed on a variety of tissues and species to catalog cell types and states[8–16], identify genomic features that activate at specific steps along cellular trajectories[17–23], and infer regulatory networks cataloging interactions of genes, open chromatin regions or methylation sites[24–26]. A common step of single cell data analysis is cell state inference: the inference of a low dimensional representation of a single cell data modality, that is subsequently used for 2D data visualization[27,28], clustering to identify discrete cell types and states, and trajectory inference tasks[29,30].

More recently, multimodal assays that can profile two or more data modalities from the same cell have been developed. For example, the Paired-seq[31], SHARE-seq[32], SNARE-seq[33], and 10X scMultiome assays simultaneously profile RNA and accessible chromatin regions, while the Patch-seq assay jointly measures cellular imaging, patch clamp electrophysiology and RNA sequencing of single neurons[34,35]. Paired multimodal assays are needed for tasks related to establishing patterns of covariation between data modalities, such as identifying chromatin regions whose accessibility is correlated with gene expression patterns, which in turn suggests the location of active regulatory elements. More broadly, paired multimodal assays help generate maps between cell state spaces derived from different data modalities, which help establish holistic views of cell state[36–38]. For example, identification of the number of distinct cell states and cell types using scRNA-seq studies alone is challenging, in part because choosing the number of clusters of cells is either an implicitly or explicitly defined parameter[39]. Recent Patch-seq studies of mammalian neurons have shown that there is a lack of statistically significant differences in electrophysiological response patterns between some transcriptionally distinct cell clusters[35,40], suggesting using a secondary data modality can help determine whether two clusters of cells should be considered distinct or not.

Several computational methods are available to map between cell states defined by different single cell data modalities[41,42]. However, these approaches have two key limitations. First, these methods all employ feature selection as an initial preprocessing step[29] that is performed independently for each data modality and is typically based on

[1]Integrative Genetics and Genomics Graduate Group, University of California, Davis, CA, USA. [2]Genome Center, University of California, Davis, CA, USA. [3]Department of Molecular and Cellular Biology, University of California, Davis, CA, USA. ✉e-mail: hrhu@ucdavis.edu; gquon@ucdavis.edu

feature variance alone. They therefore do not take into account whether the selected features are broadly correlated with any features from the other data modalities. The more features selected for one data modality that are uncorrelated with features of another modality, the worse the cell state mapping performance is expected. Second, some prevailing computational multimodal methods[43,44] rely on multimodal datasets to define mapping between data modalities with matched, or anchored, features and cells. Although emerging diagonal integration methods tend to impose more relaxed constraints, achieving accurate diagonal integration is challenging due to the lack of prior knowledge of how the same cell type presenting in one feature space maps to the same cell type in a different feature space[45]. Furthermore, multimodal datasets frequently exhibit shallow effective sequencing depth (e.g. the average number of unique molecular identifiers mapped to a cell) and lower throughput relative to unimodal datasets of the same cell populations[32-35,46], leading to a reduction in statistical power to map cell states between distinct data modalities.

Here we present scPair, a deep learning framework for computational analysis of single cell multimodal data. scPair performs automatic, implicit feature selection to infer the subset of features of each data modality that yield optimal mappings of cell states between data modalities. Our training procedure of scPair also addresses challenges of shallow sequencing of paired multimodal datasets by using unimodal data sequenced at higher effective depth to learn robust covariance structure in each data modality, and in turn relies on multimodal data primarily for implicit feature selection and mapping of cell states between data modalities. We demonstrate that these two properties of scPair enable it to outperform existing methods on multimodal data analysis tasks such as cell state mapping and feature prediction, as well as simultaneous trajectory inference in both RNA and ATAC data components to identify time point-specific feature activity during cellular differentiation.

## Results

### scPair is a deep learning framework for single cell multiomics analysis

scPair is a supervised learning framework that leverages single cell multimodal datasets to (1) infer cell states in each data modality via dimensionality reduction, (2) infer mappings between cell states defined by different data modalities, and (3) predict individual features and cell states of one data modality, given data from another data modality. scPair is composed of a pair of feedforward networks (FFN) that individually accept one data modality's features as input, and predict another data modality's features as output (Fig. 1a). The last layer of each FFN, which encodes what we term the cell state space, represents a non-linear combination of input features that are most predictive of the other data modality's features.

What distinguishes scPair's cell state space from the latent space of other methods[41,47-49] is that it is computed by implicitly selecting a subset of input features (from the entire set) that maximally predict the other data modality. In contrast, other approaches rely on an initial feature selection step that can remove up to 90% of RNA input features or 75% of ATAC input features before analysis[50-52], and is typically performed independently for each data modality. This both increases the chances of removing features that are informative of the other data modality, as well as increases the chances of including features that are irrelevant for mapping to the other data modality. For scPair, we input all features into the scPair framework, and allow scPair to automatically identify features useful for mapping to the other data modality during training. For example, in the RNA FFN in Fig. 1a, the RNA cell state layer is a low dimensional representation of genes that are predictive of many ATAC features (and therefore are more likely to be mappable to the ATAC cell state space). This gives rise to the property of scPair that we refer to as implicit feature selection: feature selection is implicitly performed by scPair when it is trained to predict all ATAC

features based only on RNA features. The two cell state layers are directly connected via bidirectional feedforward networks that serve as cell state mapping functions (Fig. 1).

During the training of scPair, the parameters of the pair of FFNs are first separately optimized by training individual FFNs to predict one data modality's features from the other's. Then, the parameters of the bidirectional mapping networks connecting the cell state layers of the encoding FFNs are updated to maximize accuracy of predicting the cell state of one modality based on the cell state defined by the other modality (Fig. 1, Methods, and Fig. S1).

### scPair robustly maps cell state representations between RNA and ATAC data

Genomic assays such as scRNA-seq and scATAC-seq measure complementary features of a cell's state and function[53]. Since multimodal single cell assays measure feature sets such as RNA and ATAC on the same cell, we intuitively expect that there should exist a mapping between RNA-based cell state and ATAC-based cell state. We therefore first evaluated scPair based on how accurately it can predict the correct ATAC-based cell state using only its RNA cell state, and vice versa.

We assembled a benchmark consisting of seven gold-standard multiomic (paired single cell RNA-seq and ATAC-seq) datasets spanning multiple tissues and species[32,33,47,54-56], and used them to benchmark scPair and three other methods that are based on widely used generative variational autoencoders (VAE)[57]: Polarbear[41], MultiVI[48], and Cobolt[49]. MultiVI and Cobolt are unsupervised methods that learn both joint and modality-specific embeddings to facilitate cell state mapping. To mitigate a technical issue running Polarbear, we implemented a semi-supervised framework termed Polarbear* that extends Polarbear[41] (Methods). To evaluate the methods, we partitioned the cells in each dataset into mutually exclusive training and testing sets. The methods were provided with the paired RNA and ATAC training data to learn cell state mappings. After training, the methods were given only the RNA portion of the held-out test cells and tasked with predicting the corresponding ATAC cell state. This process was then repeated by giving the trained methods only the held-out ATAC data and evaluating their RNA state predictions.

Figure 2 compares the predicted and ground truth RNA and ATAC cell states, respectively, on each of the seven benchmark datasets. scPair accurately mapped from RNA→ATAC (Fig. 2a) and ATAC→RNA (Fig. 2b), achieving performance competitive with the state-of-the-art (SOTA) MultiVI (with modality penalty term during optimization), and outperforming other methods by an average of 11.71% for RNA→ATAC (Fig. 2a, c, and S2a), and 19.47% for ATAC→RNA (Fig. 2b, d, and S2a). Overall, benchmarked methods other than scPair demonstrated a bias towards higher RNA→ATAC mapping accuracy compared to ATAC→RNA mapping accuracy (Fig. S2a) (4.6% difference on average, p-value = 0.02, paired Wilcoxon test). In contrast, scPair yielded low, not statistically significant bias in mapping accuracy (0.84% difference, p-value = 0.16, paired Wilcoxon test) while achieving the highest overall mapping accuracy, suggesting our strategy of implicit feature selection helps maximize mapping performance for both modalities. As mentioned above, ATAC→RNA mapping accuracies were generally lower than RNA→ ATAC mapping accuracy across methods; we attribute this poorer performance to the poorer separation of cell types in ATAC space. In our seven benchmark datasets, we observed that ATAC cell state spaces generally exhibited poorer clustering of cell types compared to RNA cell state spaces (Fig. S2b), consistent with previous observations[33]. Notably, scPair demonstrated the highest consistency between the clustering accuracy of the two modality-specific cell state spaces (Pearson correlation coefficient = 0.97).

Part of the motivation of the design of scPair is to circumvent the need to perform feature selection on the input data modalities before training. We directly tested the impact of feature selection on scPair by

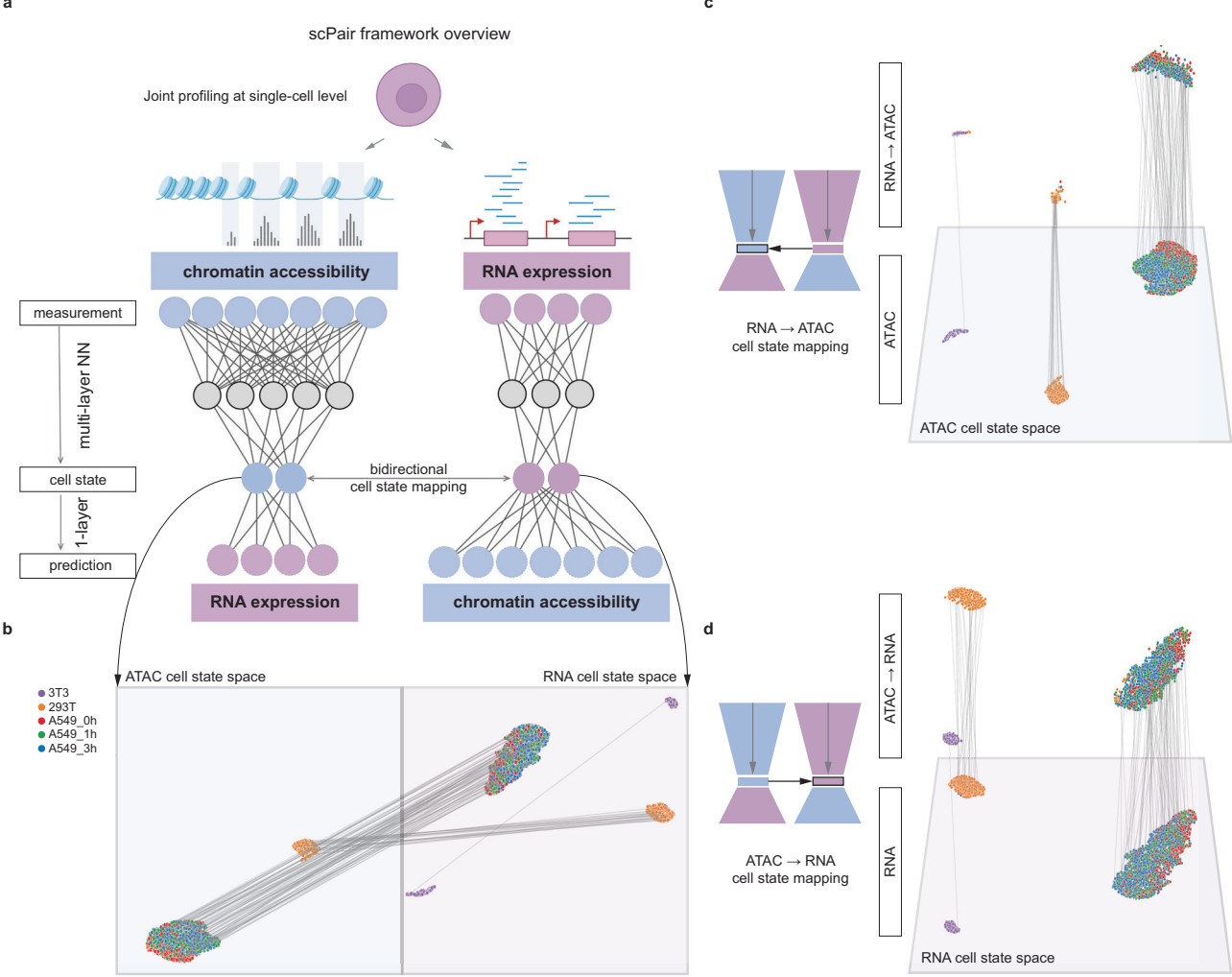

**Fig. 1 | Overview of the scPair framework for single cell multimodal analysis.**
**a** scPair uses dual feedforward neural networks to predict each modality from the other. The last hidden layer of each network encodes a modality-specific cell state space, and the bidirectional networks learn mappings between the modality-specific state spaces. (Cartoons of the single cell and assays were created with BioRender.com: Created in BioRender. Hu, H. (2024) BioRender.com/r97r180). **b** We use UMAP to visualize modality-specific cell state spaces learned by scPair. In this figure, the data is from the sci-CAR multimodal cell line dataset[55], where cells are colored by the cell type labels from the original study. Lines connect the

modality-specific states of the same cell. **c** A visualization of the bidirectional map trained by scPair. Given a multimodal single cell sample, scPair is in part evaluated based on how well it can predict the ground truth (measured) ATAC cell state (bottom), given only the RNA profile to predict the ATAC state of a cell (top). Lines connect each cell's predicted ATAC cell state to its ground truth ATAC cell state; vertical lines indicate high prediction accuracy. **d** Same as (**c**), but visualizing the ground truth (measured) RNA cell state (bottom) and the predicted RNA state from ATAC (top). Source data are provided as a Source Data file.

training a version of scPair in which we perform variance-based feature selection before scPair training, and observed a decrease of 21% in mapping performance on average (from 0.92 – 0.73) when performing feature selection ahead of time ($p$-value = $1.22 \times 10^{-4}$, paired Wilcoxon test, Fig. S3). We also confirmed that despite using many more features as input compared to other approaches, scPair is still fast to train: on our benchmarked datasets, scPair was the fastest method to train in the dataset with the most number of cells and most number of ATAC features (Fig. S4). scPair therefore overall achieves state-of-the-art performance and is fast to train.

### scPair cell state mapping enables accurate prediction of individual data features

Besides capturing global patterns of covariation between RNA and ATAC cell state spaces, another goal of collecting multimodal single cell data is to identify patterns of covariation between individual features across data modalities[58]; for example, to identify individual chromatin regions whose accessibility is correlated with local gene

expression patterns, suggesting regulatory activity. Intuitively, methods that can map between global RNA and ATAC cell state spaces should also yield accurate cross-modality prediction of individual features. We therefore assessed scPair's accuracy in predicting individual gene expression levels from the paired chromatin accessibility patterns and vice versa, compared to the other methods.

Figure 3a, b illustrate the prediction performance ranking of scPair and other methods on the held-out datasets for predicting RNA from ATAC (Fig. 3a) and predicting ATAC from RNA (Fig. 3b) at the individual feature level. Overall, scPair ranks highest in both cross-modality prediction tasks (Fig. 3a, b, Source Data); note we did not include Cobolt in these comparisons because it does not output ATAC predictions as binary profiles as the other methods do, making it challenging to compare Cobolt to other methods. We have further benchmarked scPair against scMDC[59], which concatenates features from both modalities before encoding them into the cell state space. scPair outperforms scMDC on 6 out of 7 benchmark datasets (Source

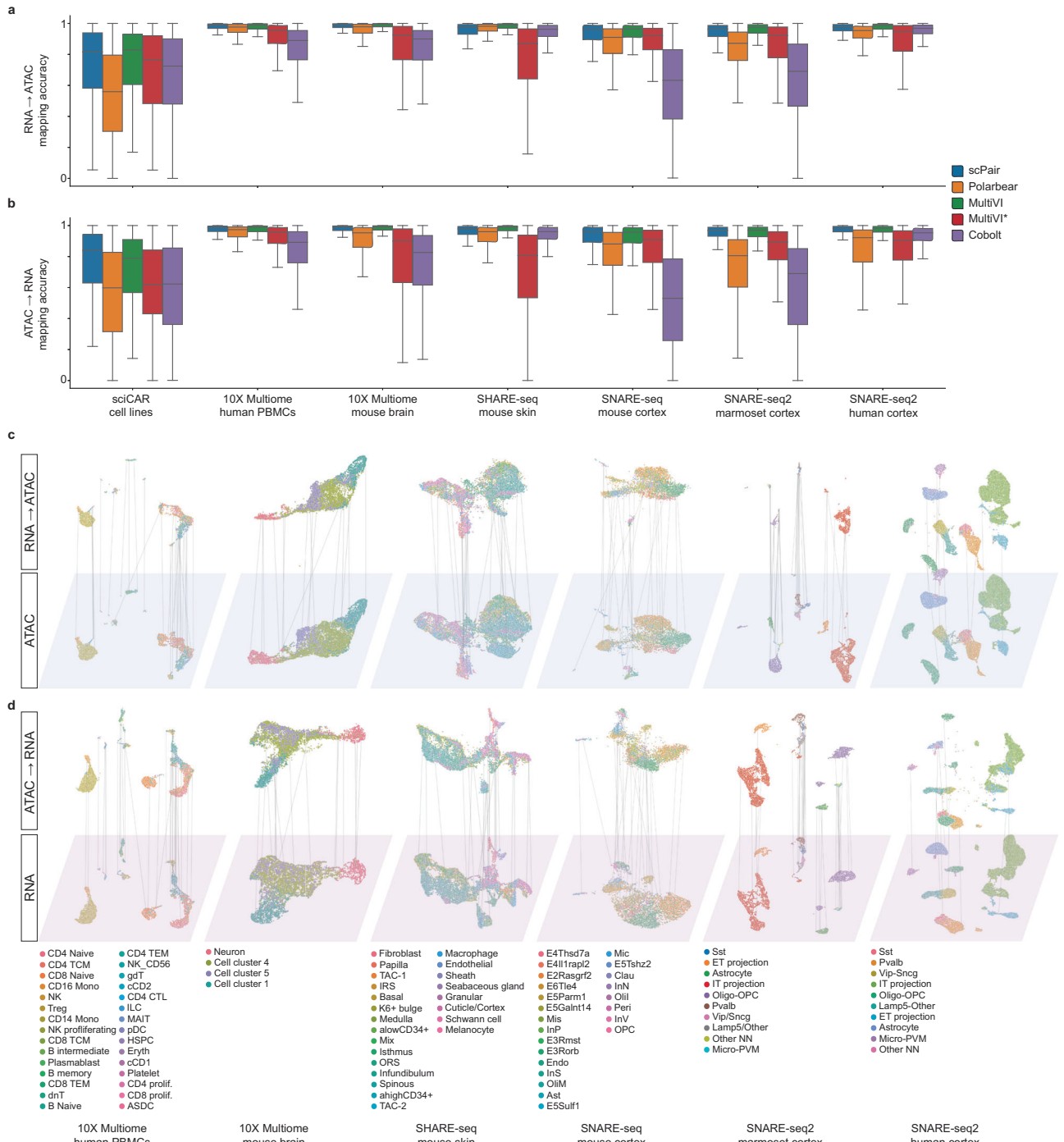

**Fig. 2 | scPair robustly aligns single cell multiomic data modalities.**
**a** Benchmark of RNA→ATAC mapping performance of scPair and other single cell multiomic methods. All methods were provided with the same training and held-out data sets for evaluation. Box plots compare the mapping performance as measured by the Fraction Of Samples Closer Than the True Match metric (1-FOSCTTM), where larger values indicate better performance. In the box plots, the minima, maxima, centerline, bounds of box, and whiskers represent the minimum value in the data, maximum, median, upper and lower quartiles, and 1.5x inter-quartile range, respectively. **b** Same as (**a**), except measuring ATAC→RNA

performance of all methods. **c** UMAP visualizations of the ATAC (ground truth) and RNA→ATAC (predicted) cell state spaces learned by scPair on single cell multiomic datasets. Each point represents a single cell, and lines connect each cell's measured ATAC and predicted ATAC (via mapping RNA→ATAC) cell states. Colors correspond to cell type labels from the original studies[32,33,54–56] (datasets from left to right: 10X Genomics scMultiome human PBMCs, 10X Genomics scMultiome mouse brain, SHARE-seq mouse skin, and multi-species SNARE-seq cortex datasets). **d** Same as (**a**), but visualizing the RNA (ground truth) and ATAC→RNA (predicted) cell states learned by scPair. Source data are provided as a Source Data file.

Data). However, scMDC suffers from label leakage, as it requires both RNA and ATAC inputs for predicting RNA, making it unsuitable for cross-modal prediction tasks. In contrast, scPair, Cobolt, Polarbear, and MultiVI can independently predict one modality from the other, making their comparison more fair.

Figures 3c and d compare the predicted and ground truth RNA profiles of held-out cells from the mouse cortex SNARE-seq dataset. scPair predictions accurately capture cell type-specific gene expression patterns, with individual cells of a given type predicted to express key known markers of their respective cell type. This

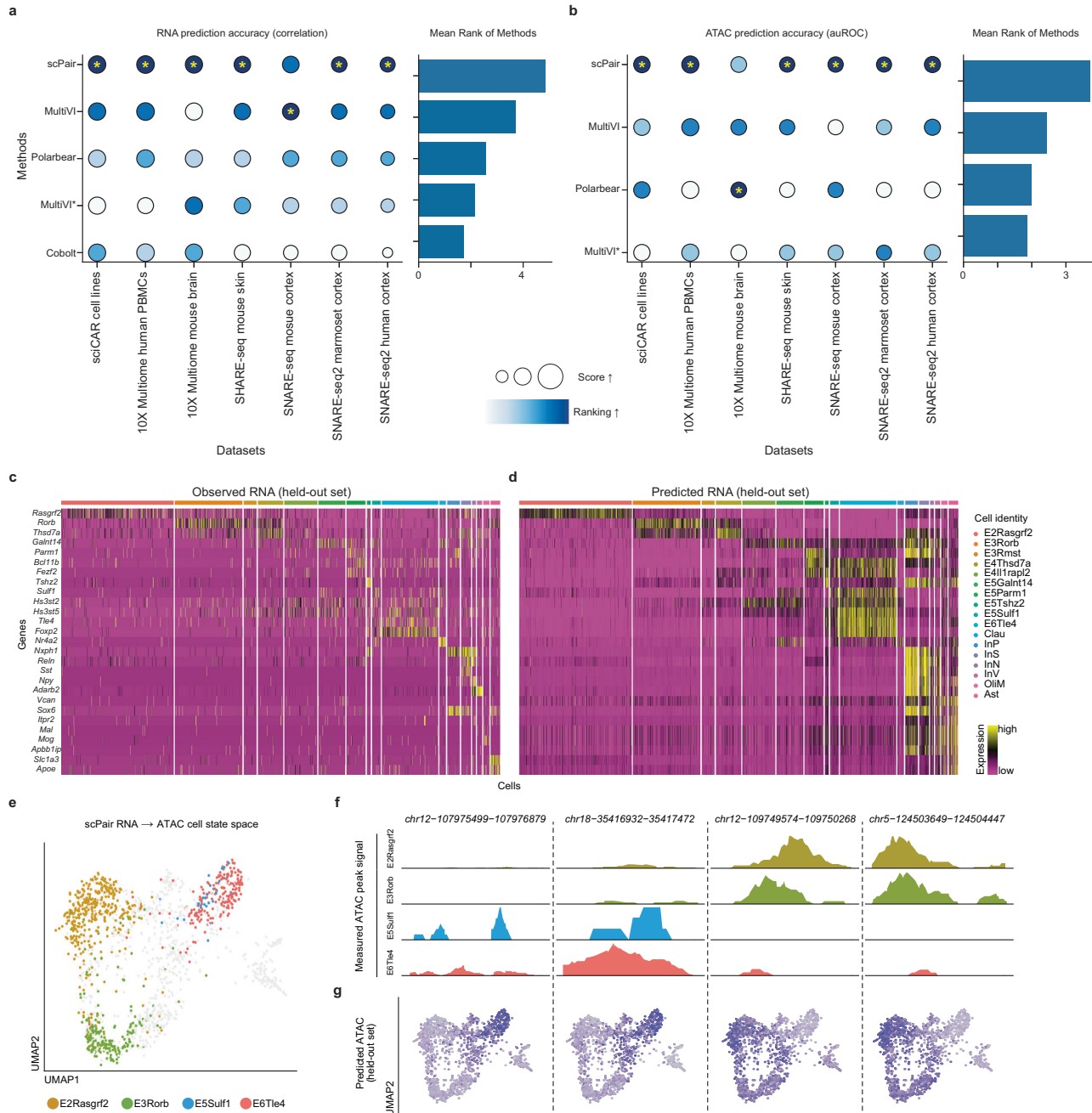

**Fig. 3 | Prediction of individual data features from the other data modality.**
**a** The ranking of RNA expression prediction accuracy, measured as Pearson correlation, across held-out data from seven datasets. Yellow stars indicate the best performing methods. **b** Same as (**a**), except the ranking of ATAC opening prediction accuracy, measured as area under the Receiver Operating Characteristic curve (auROC). **c** Held-out ground truth RNA expression from each cell type from the SNARE-seq multimodal adult mouse cortex dataset. Rows are differentially expressed genes and columns are cells clustered by type. **d** Predicted RNA expression based on the held-out ATAC profiles from the SNARE-seq multimodal adult mouse cortex dataset. Rows are differentially expressed genes and columns

are cells clustered by type, in the same order as (**c**). **e** UMAP of scPair's predicted ATAC cell state space (based solely on the RNA measurement of held-out samples), where cells are colored by cell types that have been defined in the SNARE-seq multimodal adult mouse cortex dataset. **f** Aggregated held-out ground truth accessibility tracks for the example marker peaks, which are identified as those that differ between cortical layer 2-3 (E2Rasgrf2, E3Rorb) and layer 5-6 (E5Sulf1, E6Tie4) excitatory neurons, within each corresponding cell type shown in (**e**). **g** UMAPs showing the predicted accessibility of peaks in (**f**), based on held-out RNA profiles. Color indicates opening probability. Source data are provided as a Source Data file.

suggests that the RNA cell state space predicted by scPair from the ATAC cell state space accurately captures global gene co-expression patterns.

As an additional set of visualizations to evaluate the quality of predicted per-chromatin region accessibility profiles, Fig. 3e illustrates a UMAP visualization of the predicted single cell chromatin accessibility cell state derived from RNA measurements of held-out test cells

in the SNARE-seq mouse cortex dataset. These cells are labeled by the predefined cell type from the original study[33], and include cell types such as cortical layer 2-3 and layer 5-6 excitatory neurons, among others. Globally, the predicted ATAC cell states successfully capture the separation of cell types in these held-out cells not used for training, suggesting that scPair effectively projects RNA to ATAC cell states. At the individual locus level, we identified four chromatin regions whose

accessibility patterns were identified as markers of cell types (Fig. 3f); we see their predicted accessibility patterns in the held-out samples (Fig. 3g) are consistent with their corresponding cell types (Fig. 3e) (excitatory neuron layer 2-3 and layer 5-6 subtypes). These results support the robustness of scPair in predicting individual chromatin region accessibility patterns accurately, using only the paired scRNA-seq data.

## Simultaneous trajectory inference across RNA and ATAC cell state spaces

Trajectory inference is a powerful technique for identifying genes and chromatin regions that exhibit dynamic, context-dependent or time-dependent activity patterns, and is typically applied to low-dimensional (cell state) representations of cells. Historically, it has been applied most extensively on scRNA-seq data; applications to scATAC-seq data have been challenging because of the noisy cell state spaces inferred for ATAC data compared to RNA[60]. Here we explored whether we could use scPair-inferred RNA and ATAC cell states as input to existing trajectory inference methods to identify regulatory elements with roles in cell differentiation.

We first obtained a neonatal mouse brain SNARE-seq dataset of 1469 cells representing five cell types spanning cortical development at the postnatal day 0 (P0) stage[33], and trained all models on this dataset. Unlike the adult mouse SNARE-seq data analyzed in Figs. 2 and 3, these cells are expected to follow a linear developmental trajectory from intermediate progenitor IP-Hmgn2 to excitatory layer 2/3 Cux1+ neurons. This dataset is challenging in part because of the presence of significant batch effects (Fig. S5a, b). Figure 4a illustrates how while most methods infer RNA cell state spaces that accurately separate cell types, only MultiVI[48] and scPair are also able to infer ATAC cell state spaces that separate cell types. We therefore reasoned it was sensible to explore trajectory inference only on the scPair and MultiVI cell state spaces.

We first performed trajectory inference using diffusion maps[61–63] on the RNA and ATAC spaces separately for each method. Each cell was therefore assigned two pseudotimes, one based on its RNA measurement, and one based on its ATAC measurement. Both MultiVI and scPair yield the most consistent pseudotime assignments of cells between the RNA and ATAC measurements (Fig. S5c), indicating cells are similarly arranged in the RNA and ATAC spaces for both methods. However, scPair additionally achieves high consistency and accuracy in clustering of cell types for both RNA and ATAC, while MultiVI achieves high consistency but poor accuracy in clustering of cell types (Fig. S5c). In other words, MultiVI yields high consistency between RNA and ATAC spaces likely due to its modality penalty that encourages this consistency, but this consistency comes at the cost of the quality of clustering of cell types in RNA space as seen by comparing MultiVI to MultiVI* in Fig. 4a. As a result, scPair outperforms MultiVI in terms of ordering the five cell types in their established developmental stage in the trajectory (Fig. 4b, S5d). scPair also successfully recovered the coordinated expression changes of canonical marker genes along the cortical maturation trajectory compared to the other approaches (Fig. 4c, Fig. S5e, f, and Fig. S6). These results demonstrate how scPair helps improve trajectory inference in both RNA and ATAC cell state spaces.

## Unimodal datasets increase the density of cells along multimodal trajectories

Two key limitations of current multiomic assays compared to unimodal assays is their relatively small sample size (Fig. S7a), likely due to the increased experimental complexity and cost[53,64], and shallow sequencing depth[32,41] (Fig. S7b, c). Smaller sample sizes and shallow sequencing together are expected to yield more difficult trajectory analysis due to poorer coverage of genes and chromatin regions specifically expressed during transient cell states.

We reasoned that integrating these small, shallow multimodal datasets with larger, higher depth unimodal datasets can help to increase the effective sequencing coverage of transient cell populations in the trajectory. To test this hypothesis, we compare two strategies for leveraging scPair to perform trajectory analysis. Here, we assume we have a (smaller) multimodal RNA and ATAC dataset, as well as a larger unimodal scATAC dataset. In both strategies, we initially train scPair on the multimodal data only. Then, in the first strategy, we subsequently pass the RNA and ATAC-based cell states estimated for the multimodal data only to an existing trajectory inference method[62] to estimate pseudotimes of each modality separately, yielding the pseudotime estimates scMultiome$_{RNA}$ and scMultiome$_{ATAC}$. This strategy represents data analysis using only the smaller, multimodal dataset. The second integrated strategy leverages a larger unimodal scATAC-seq dataset by estimating ATAC cell states using the trained scPair model, and passing them through the trajectory inference framework[62] to assign each unimodal scATAC-seq cell their scUnimodal$_{ATAC}$ pseudotime (Fig. S1d). In both cases, trajectory inference is used to estimate pseudotime of input cells, and thus infer transiently active genes and chromatin regions; however, the second integrated strategy augments analysis of the multimodal dataset with the larger unimodal scATAC-seq dataset, and the first strategy relies only on the smaller multimodal dataset. We presume the strategy that yields more transiently active genes and regions is more informative of transient states during trajectory inference. More details can be found in the Supplementary Information and Methods section.

To demonstrate this strategy, we obtained data from a multiomic study by Allaway et al.[65] that sequenced differentiating neuron precursor cells in order to determine the earliest precursor states in which interneurons can be distinguished from projection neurons. At mouse embryonic day 13 (E13), a key stage for examining transcriptional and epigenetic dynamics in the medial ganglionic eminence (MGE)[65,66], 14,605 unimodal scATAC-seq and 2,141 scMultiome postmitotic cells have been profiled and passed quality control (Methods).

In the Allaway study, part of their goal involved ultimately assigning pseudotimes to the unimodal scATAC-seq dataset. To do so, they implemented a multi-step label transfer process involving first using a unimodal scRNA-seq dataset to infer developmental trajectories and applied label transfer to assign pseudotimes to the paired multimodal data. The sparse multiomic data was then used as a bridge to further transfer trajectory pseudotime labels to the unimodal scATAC-seq cells. Here we avoided the multi-step label transfer process and directly assign pseudotimes to the unimodal scATAC-seq dataset using our scPair strategy outlined above.

We initially trained scPair on 2141 postmitotic 10x scMultiome cells and performed trajectory analysis on those same multimodal cells to estimate their pseudotimes (scMultiome$_{RNA}$ and scMultiome$_{ATAC}$, Fig. 5a and Fig. S8a). We then executed our integrated strategy to assign scUnimodal$_{ATAC}$ pseudotimes to the 14,605 unimodal postmitotic scATAC-seq samples. Figure 5b and S8b illustrate how the 14,605 unimodal scATAC-seq samples and assigned scUnimodal$_{ATAC}$ pseudotimes support a trajectory with three terminals and branches, preserving the unimodal RNA-based trajectory structure seen in the original study[65]. For comparison, we also illustrate that a similar trajectory is obtained by only looking at the ATAC component of the 2,141 10x scMultiome cells and their associated scMultiome$_{ATAC}$ pseudotime, though there is clearly less coverage of transient states (Figs. 5a, S8a) compared to using the unimodal scATAC-seq dataset (Fig. 5b, S8b) due to the 7-fold more cells profiled by the unimodal scATAC-seq (Fig. S7a). Using scPair, we predicted the RNA expression profiles of each of the 14,605 unimodal scATAC-seq samples, and confirmed that known markers of cell states along the separate trajectory branches were recapitulated (Fig. 5c, d and Fig. S8c, d). For instance, *Fabp7* marked the initial progenitor state, while *Maf*, *Zic1*, and *Ebf1* specified terminal states of the three trajectory branches (Fig. 5c,d).

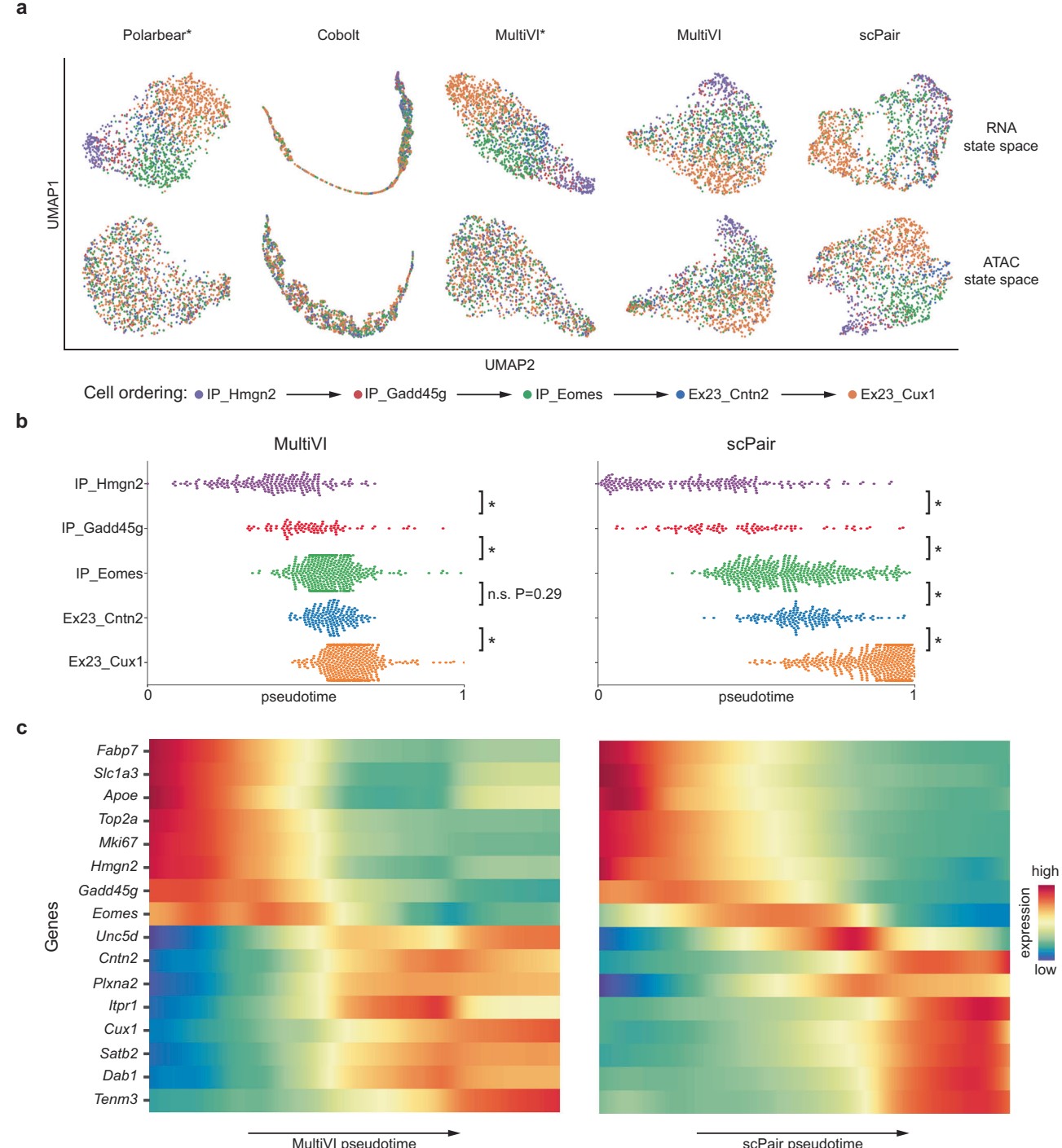

**Fig. 4 | Inference of developmental trajectories in ATAC space. a** UMAP visualizations of cell state spaces learned from RNA (top) and ATAC (bottom) by various methods on the neonatal mouse cortex SNARE-seq data. Colors indicate cell types as defined in the original study. Below, a diagram indicating the expected linear developmental trajectory[33]. **b** Swarm plots illustrate the individual pseudotimes assigned to each cell of each cell type, which are inferred using the cell state spaces learned by MultiVI (left) and scPair (right). The order of cell type on the y-axis (from top to bottom) follows the developmental path observed in the original study. * represents significance (*p*-value < 0.05) using a two-sided *t*-test, and n.s. represents non-significance. **c** Heatmaps of developmental state marker expression along pseudotime (*x*-axis) inferred from the MultiVI (left) and scPair (right) ATAC spaces. Gene order on the *y*-axis follows the expected order of expression according to maturation time from the original study. Source data are provided as a Source Data file.

Given the larger size of the unimodal scATAC-seq dataset, we next investigated whether trajectory analysis of the unimodal scATAC-seq data could identify new transiently open chromatin regions that are accessible only along specific differentiation trajectories. Figure 5e visualizes the pseudobulk chromosome accessibility patterns as a function of either the scUnimodal_ATAC or the scMultiome_ATAC estimated pseudotime, corresponding to the 14,605 unimodal scATAC-seq samples and the 2,141 10x scMultiome samples, respectively (Methods). We found that the increased size of the unimodal scATAC-seq dataset provided greater power to detect transiently open chromatin regions in different branches of the trajectory, as evidenced by stronger

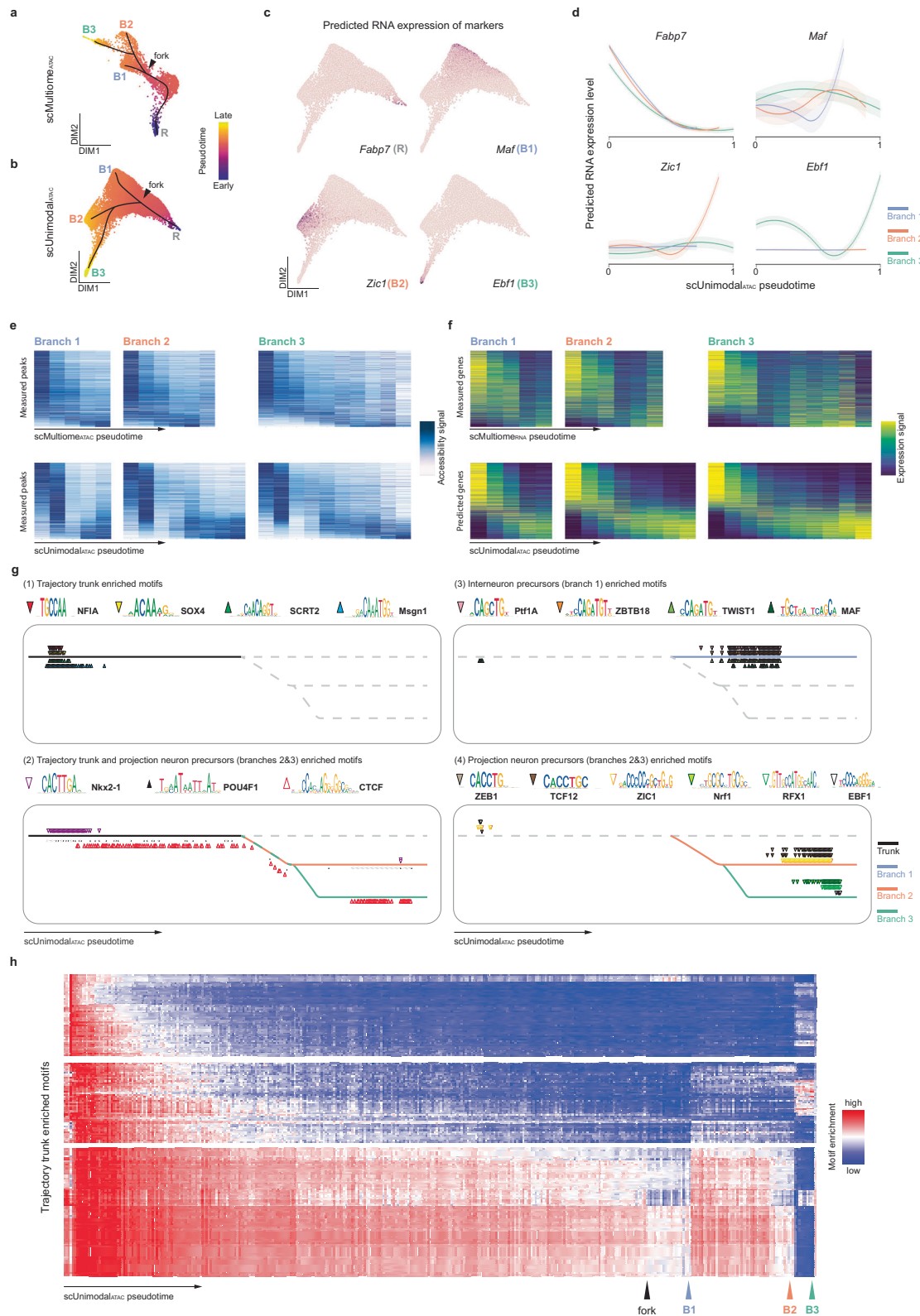

pseudotime-specific accessibility patterns for the unimodal scATAC-seq based pseudobulk samples compared to the 10x scMultiome-based pseudobulk samples (Fig. 5e). For instance, between the formation of the initial branch and the end of the trajectory, the unimodal scATAC-seq data identifies 2.8-fold more pseudotime-specific peaks compared to 10x scMultiome data (5217 versus 1854).

To determine the extent to which these transiently active chromatin accessible regions translated into changes in gene expression patterns, we used scPair to predict RNA expression patterns for the 14,605 unimodal scATAC-seq cells, and compared trajectories of 10x scMultiome RNA cell states to trajectories of the predicted RNA cell states of the unimodal scATAC-seq cells (Fig. 5f). Using the unimodal scATAC-seq data, we identified 2.2-fold more pseudotime-specific

**Fig. 5 | Augmenting multiomic analysis with unimodal datasets enhances coverage of transient states in trajectory inference. a-b** UMAP visualizations of the ATAC cell state spaces learned by scPair, with 2,141 10x scMultiome cells used to train scPair (**a**), or with the 14,605 unimodal scATAC-seq cells (**b**). Cells are colored based on estimated pseudotime trajectories via Palantir, with labels (R, B1, B2, B3) indicating the trajectory root and branch terminals. Arrows mark the initial fork points. **c** UMAP visualizations of unimodal scATAC-seq cell state learned by scPair, each corresponding to the predicted expression pattern of specific marker genes: *Fabp7* (starting pluripotent state), *Maf*, *Zic1*, and *Ebf1* (markers of branch 1, 2 and 3, respectively). **d** Line plots show RNA expression predicted by scPair from the unimodal scATAC-seq data for the four markers from (**c**), as a function of pseudotime. Error bands represent one standard deviation. **e** Heatmaps compare chromatin accessibility patterns along inferred pseudotime (*x*-axis) for each branch in the trajectory, using the 2,141 10x scMultiome cells (top) versus the 14,605 unimodal scATAC-seq cells (bottom). Rows represent features (peaks), and

columns represent 0.05 pseudotime intervals. In each heatmap, the order of rows from top to bottom is based on "feature pseudotime" (Methods) in ascending order. **f** Same as (**d**), except comparing measured RNA expression from the 10x scMultiome cells (top) and predicted RNA expression by scPair using the unimodal scATAC-seq cells (bottom). **g** Pseudotime-specific enrichments of transcription factor binding motifs along trajectory. Motifs found to be enriched in accessible regions of transient states were categorized as either (1) enriched in the trajectory trunk; (2) enriched in both trunk and projection neuron precursor branches; (3) mainly in branch 1 corresponding to interneuron precursors; and (4) projection neuron precursor branches only (branches 2 and 3). Example motifs were selected for visualization for each of the four categories. **h** Heatmap displays motif enrichment along trajectory, with vertical arrows marking the fork and branch terminals indicated in (**b**). Rows represent the enriched motifs and columns represent pseudotime. Source data are provided as a Source Data file.

gene expression patterns comparing to when using the 10x scMultiome RNA measurements (2890 versus 1296). Because our comparison was between two different uses of scPair, the improved identification of branch-specific genes is driven specifically by our integration of the larger unimodal scATAC dataset, rather than being driven more generally by the use of scPair to predict RNA states of the unimodal scATAC-seq data. To further demonstrate this, we repeated our above analysis except we only used the ATAC component of the 10x scMultiome cells to predict RNA profiles and infer pseudotimes (instead of the larger unimodal scATAC-seq dataset). As shown in Fig. S8e, the results of this 10x scMultiome-based analysis do not capture branch-specific gene expression trends as well as when we use the larger unimodal scATAC-seq dataset (Fig. 5f, bottom), suggesting our improved identification of branch-specific genes is driven primarily by the use of a larger unimodal scATAC-seq dataset.

Given we were able to identify more transient, accessible regions along different branches using the scATAC-seq data, we reasoned we will have more power to detect transcription factor binding motifs enriched in specific differentiation states. The presence of cell state-specific motifs and opening chromatin regions would suggest regulatory factors involved at key points during differentiation. To identify key regulators that may play a role in driving differentiation and cell fate decisions, we performed a motif enrichment analysis on chromatin regions exhibiting transient accessibility along pseudotime trajectories for the early-stage trunk and all three branches (Fig. 5g).

We tested 841 motifs for enrichment in chromatin regions using ChromVAR[67], and discovered transcription factors (TFs) with binding motifs highly enriched in a branch specific manner. We distinguished motifs based on their primary trajectory location of enrichment: (1) 186 motifs enriched at the trunk of the differentiation trajectory but rarely at the end of any branch terminals; (2) several motifs enriched along both the trajectory trunk, as well as in the projection neuron precursors (branches 2 and 3); (3) 30 motifs mainly enriched in the interneuron precursors (branch 1); and (4) 76 motifs in total enriched principally in the projection neuron precursors (branches 2 and 3). Figure 5g highlights several enriched motifs. These include NFI and SOX family genes that coordinate and play critical roles in early neuronal progenitor differentiation and brain development[68–72]. Many of these factors have conserved regulatory roles in the embryonic medial ganglionic eminence (MGE) between human, non-human primate (NHP), and mouse[70,73]. Interestingly, Allaway et al.[65] identified Nkx2-1 as a specific regulator in branch 2. Our results point to potential dual-stage roles of Nkx2-1, both in early differentiation before commitment to any branch, as well as in the late stage of branch 2 (Fig. 5g). We identified several other motifs enriched in multiple stages (Fig. 5g (**2**)), including the POU4F class and CTCF that exhibit enrichment along the trajectory trunk and projection neuron precursors terminal but not enriched in branch 1. These TFs have established roles in either axon

projection and guidance or are required for cortical neuron distribution across cortex layers[71,74–76]. Furthermore, among the motifs enriched in the trajectory trunk, we also identified three major groups of motifs that display varying activity patterns along pseudotime (Fig. 5h). One group of motifs (52 motifs) shows activity specifically around early pseudotime stage before rapidly declining. A second group of motifs (51 motifs) exhibits extended activity before branch forking. The last group of motifs (82 motifs) displays more prolonged activity, lasting longer and even extending into some branches but not terminals (**Source Data**). The different activity patterns suggest sequential and transient roles in early differentiation prior to lineage commitment, and these motif groups may function to initiate the differentiation program, consolidate neuronal identity, and prime cells for branch-specific maturation, respectively.

Together, these results suggest a novel strategy to improve the coverage of multimodal trajectory analysis using larger unimodal datasets to ultimately nominate more transient regulatory elements and genes active during trajectories.

## Unimodal datasets can mitigate the shallow per-cell sequencing depth of multimodal datasets

In the previous section, we showed how scPair can leverage unimodal datasets and mitigate the challenge of small multimodal datasets by increasing the cell density and coverage across a trajectory. Our strategy above, however, does not address the challenge of low sequencing depth of multimodal assays. We reasoned that shallow sequencing depth would also effectively reduce the number of samples available to train the encoders of scPair or other cell state inference methods. The encoders, being a form of dimensionality reduction in the case of scPair and other cell state inference methods, intuitively use the covariance structure in both the input features and non-linear transformations of them in order to reduce the dimensionality of each data modality[77–79]. However, robust covariance estimation can require many more samples (cells) compared to the number of input features, which can be a challenge since data modalities such as ATAC-seq can measure millions of features[3,13,60], and even RNA-seq can have tens of thousands of features[29]. Since multimodal data typically has lower sequencing depth and fewer cell counts than unimodal data, we hypothesized that the unimodal data might be more robust for estimating covariance of the input features deemed important during scPair training on the multimodal data. We therefore developed an augmented version of scPair that leverages unimodal data to more robustly estimate the parameters of the encoders of scPair that intuitively capture this covariance structure in the input data (Methods, Fig. S1).

We evaluated the performance of scPair without unimodal data (scPair$_{standard}$) and with unimodal data (scPair$_{augment}$) on the multiomic SNARE-seq mouse cortex data dataset with 8,055 cells, as we were able to obtain comparable unimodal scRNA-seq data[46] with 15,413 cells and

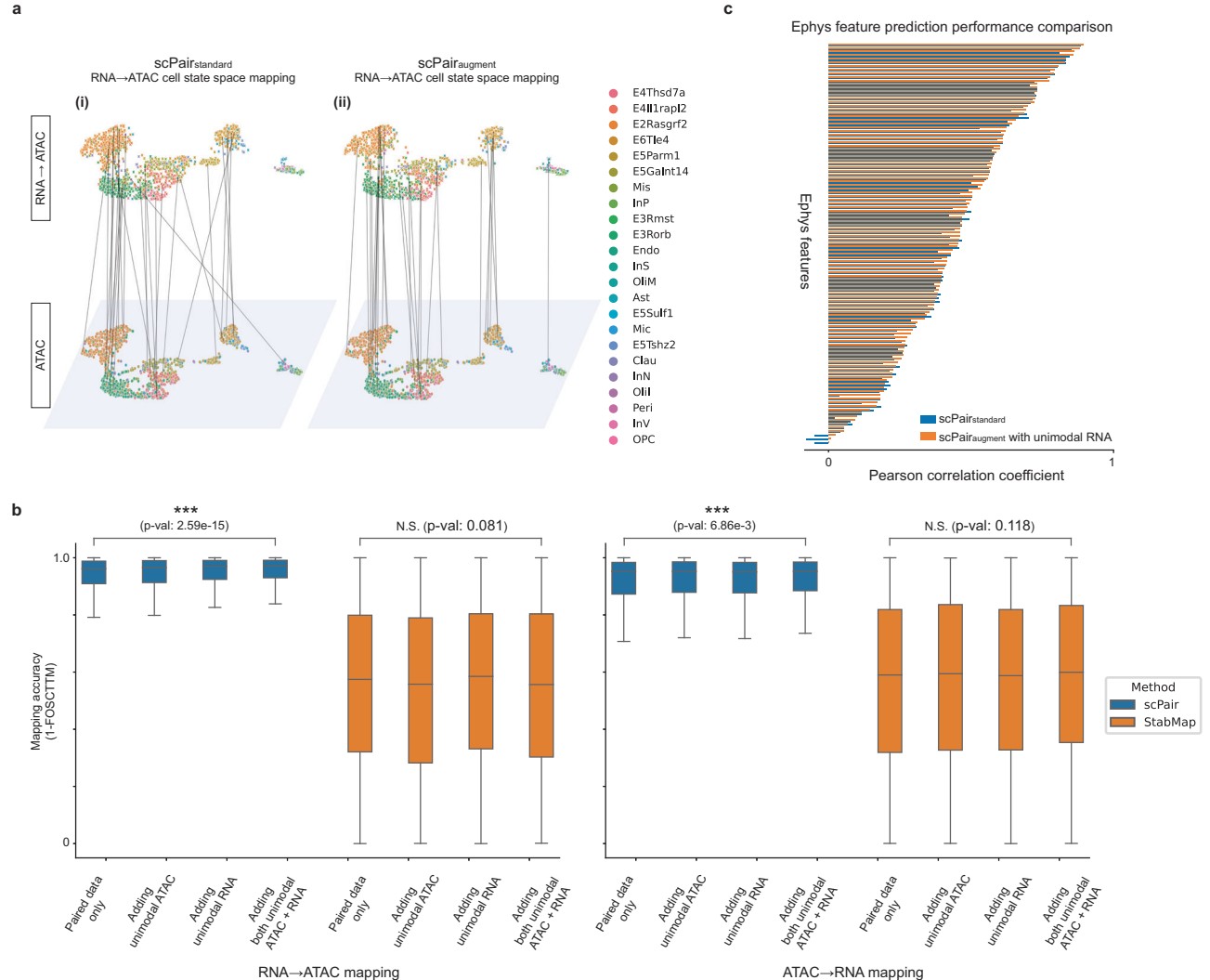

**Fig. 6 | Unimodal datasets help improve cell state inference in multimodal datasets by refining feature covariance estimation. a** UMAP visualizations comparing the accuracy of RNA→ATAC cell state space mapping by scPair, trained with (i) only the multimodal data, and (ii) after updating the scPair RNA encoding networks using a unimodal scRNA-seq atlas dataset. Each point represents a single cell, with lines connecting each cell's learned ATAC cell state and mapped RNA→ATAC cell state. Colors correspond to cell type labels from the original study. More vertical lines indicate better mapping performance. **b** Box plots quantifying the improvement in cross-modality cell state mapping (left: RNA→ATAC mapping; right: ATAC→RNA mapping) after incorporating unimodal datasets into the scPair (blue) and StabMap (orange) frameworks. Higher (1- FOSCTTM) values indicate

improved mapping performance. *P*-values from two-sided paired Wilcoxon tests indicate the significance of the improvements in cell state mapping/alignment for each method. In the box plots, the minima, maxima, centerline, bounds of box, and whiskers represent the minimum value in the data, maximum, median, upper and lower quartiles, and 1.5x interquartile range, respectively. **c** Bar plot indicating the difference in prediction performance (Pearson correlation coefficient) between the scPair$_{standard}$ framework using only bimodal Patch-seq data (blue) and the updated scPair$_{augment}$ framework incorporating a unimodal scRNA-seq atlas dataset (orange). Higher correlation demonstrates improved prediction performance. Source data are provided as a Source Data file.

comparable unimodal scATAC-seq data[13] with 28,490 cells (Fig. S7a). UMAP visualizations show alignments between the ATAC cell state space learned by scPair (Fig. 6a), and the predicted ATAC cell state from RNA using scPair$_{standard}$ or scPair$_{augment}$ (Fig. 6a). Incorporating unimodal data yields more accurate mappings between the RNA and ATAC cell state spaces, as reflected by the higher number of vertical lines connecting the same cell across state spaces (Fig. 6a), as well as by the overall higher mapping accuracy (1-FOSCTTM) and lower loss (RMSE) (Fig. 6b, S9a, **SourceData**). Also, incorporating both unimodal RNA and unimodal ATAC leads to better scPair performance compared to incorporating either unimodal RNA or unimodal ATAC data alone. This is consistent with the notion that each unimodal dataset is independently making the covariance estimation for each data modality more robust.

To further evaluate the improvement of scPair with the use of unimodal data, we compared scPair against the bridge integration method StabMap[80] that can also incorporate unimodal RNA and ATAC datasets (Methods). When mapping to RNA cell state space, the average mapping accuracy for StabMap did not significantly increase in performance after incorporating the large unimodal datasets (0.56 – 0.58, *p*-value = 0.081, paired Wilcoxon test), whereas for scPair, it improved from 0.86 – 0.89 (*p*-value = 6.86 × 10$^{-3}$, paired Wilcoxon test) (Fig. 6b). Similarly, when mapping to ATAC space, the average mapping accuracy for StabMap decreased (not significantly) from 0.55 to 0.54 (*p*-value = 0.118, paired Wilcoxon test), while for scPair, it increased from 0.93 to 0.94 (*p*-value = 2.59 × 10$^{-3}$, paired Wilcoxon test). These results also indicate that scPair overall significantly outperformed StabMap in our benchmark tests (Fig. 6b).

We reasoned one emerging area of single cell analysis that could benefit from the scPair unimodal data augmentation strategy is in the analysis of lower throughput multimodal assays, such as the Patch-seq assay. Patch-seq assays simultaneously profile morphology, electrophysiology, and transcriptomics of individual neurons, but recent studies using this assay have only managed to profile up to a few hundred cells per region and cell type[34,35], which is in stark contrast to the unimodal RNA datasets of similar regions that collected up to tens of thousands cells of the same type[46]. We therefore applied the scPair$_{standard}$ and scPair$_{augment}$ strategies to determine to what extent we could improve multimodal analysis for these small Patch-seq datasets.

We applied scPair to a Patch-seq dataset consisting of GABAergic neurons from the mouse visual cortex[35], and inferred RNA cell state spaces for both scPair$_{standard}$ and scPair$_{augment}$. We found a modest improvement in the clustering of pre-defined cell types in the RNA space with unimodal data augmentation (Fig. S9b, ARI of 0.88 compared to 0.85), likely because the RNA profiles from the multimodal Patch-seq assay were captured using full length SMART-seqv2 protocol[81] with high depth (922,330 reads per neuron on average) comparable to the unimodal scRNA-seq atlas data (1,254,047 reads per neuron on average). We also evaluated the prediction performance of individual ephys features based on RNA data (Fig. 6c and Fig. S9c, d). Incorporating unimodal data increased the average correlation between true and predicted ephys features by 4.7% (from 0.43 – 0.45 on average, $p$-value = $5.12 \times 10^{-9}$, paired sample Wilcoxon test) and decreased the average MSE loss (from 0.51 to 0.50 on average, $p$-value = $2.64 \times 10^{-8}$, paired Wilcoxon test) significantly on the held-out neuron set.

Taken together, these results demonstrate a novel way in which multimodal single cell data analysis can benefit from large collections of unimodal single cell datasets that continue to be produced today.

## Integration of multimodal single cell data beyond ATAC and RNA

The majority of our experiments thus far have focused on multimodal single cell datasets in which the ATAC and RNA modalities were measured on the same cell. We also demonstrated success on applications to Patch-seq data, showing the capability of scPair to handle diverse data types such as electrophysiological and transcriptomic profiles. Here, we briefly demonstrate scPair's ability to map between protein and RNA measurements on CITE-seq data (Fig. S10). This highlights scPair's potential application to other types of single cell multimodal datasets, including those involving RNA and Antibody-Derived Tags (ADT).

We tested scPair on a human PBMC CITE-seq[77] dataset that contains 6855 cells with 15,677 RNA features and 14 ADT features. We achieved modality-specific cell state spaces with accurate and consistent cross-modality mapping (Fig. S10a). Moreover, the clustering separation based on the cell states inferred by scPair is biologically meaningful, as shown by the high expression of the CD14 marker gene and corresponding surface protein in the same distinct cell population, which is clearly segregated from other cells with little to no CD14 expression (Fig. S10b).

These findings collectively demonstrate scPair's robustness and flexibility in integrating and analyzing diverse and complex multimodal single cell datasets, thereby providing valuable insights into cellular heterogeneity and functional states.

## Discussion

Existing methods to integrate single cell multimodal data often combine elements of autoencoders[82] or variational autoencoders (VAE)[57], in which low dimensional representations of each data modality are learned by minimizing reconstruction loss of the original input data features. While minimizing reconstruction loss of all input features is useful for learning a global representation of each input data modality, in this paper we demonstrate that to achieve high accuracy of mapping between data modalities, the low dimensional representation of each modality should be trained to predict the features of the other data modality, and use all measured features as input. Our scPair strategy can therefore be viewed as implicitly selecting features of each modality that best predict the features of the other data modality and therefore facilitate mapping. This is in contrast to existing approaches, many of which preprocess the multimodal data by using per-feature variance[41,48–50,77,83] to pre-select a subset of representative features for each data modality before low dimensional representations are learned. Variance-based feature selection is less intuitive specifically for mapping between data modalities; in the case of RNA and ATAC, there might be many accessible regions that are variable but not correlated with any genes (because they are not directly involved in gene regulation) and should therefore be selected out; the same holds true for RNA. In support of this argument, we show that variable selection as a pre-processing step based on variance leads to poorer mapping performance for scPair (Fig. S3).

Our multimodal experiments in this paper primarily focused on assays that jointly measure RNA and ATAC on single cells. In principle, scPair could be extended to accommodate other data modalities, as we showed in the case of Patch-seq which profiles the electrophysiological properties of neurons in addition to RNA (Fig. 6c and Fig. S9b-d). In addition to Patch-seq, we also demonstrated scPair's ability to map between ADT and RNA measurements made from CITE-seq data (Fig. S10). A potential future application of scPair would be to incorporate imaging or biomolecular sequence data by replacing the encoder structure with a convolutional neural network (CNN)[84] or transformer[85] architecture tailored for images or sequences. We therefore expect scPair and its associated strategies for integrating unimodal and multimodal datasets to be widely applicable for general multimodal data analysis. In contrast, there are more restricted multimodal approaches explicitly designed to map ATAC to RNA by using the ATAC data to compute a gene activity score matrix[86,87] that in principle is directly comparable to RNA data. The advantage of the gene activity score matrix is that it can be used to map scATAC-seq data to unpaired scRNA-seq data[7,50], while scPair and other approaches tested here require at least some paired (bridge) input data. However, the gene activity score approach depends on explicit prior knowledge of which non-coding regulatory regions should be grouped with which coding regions (genes), which is difficult prior knowledge to obtain genome-wide given the uncertainty over regulatory element-gene linking[88,89].

We showed two different strategies for leveraging large unimodal datasets to improve two downstream multimodal data analysis tasks: cell state mapping and trajectory inference. These strategies are useful because unimodal assays are still widely used despite the availability of multimodal assays, and even recent design studies collect more data with unimodal assays than multimodal assays[35,65,90]. Strategies for integrating both multimodal and unimodal datasets will therefore be important particularly in the study of more specialized cell types that are poorly represented in single cell experiments, or when analyzing specialized multimodal datasets such as the Patch-seq datasets analyzed here[35] for which throughput of the multimodal assay is low.

Our results show that most existing cell state mapping methods have relatively poor performance with respect to mapping cell states from the ATAC modality. There are several possible explanations: (1) the inherent sparsity and noisier nature of ATAC-seq data compared to RNA data, and (2) the fact that RNA is a more direct measurement of molecular functional activity of the cell compared to chromatin region accessibility. Even the state-of-the-art methods such as MultiVI[48] that modify the loss function to explicitly consider cross-modality mapping

accuracy, still perform relatively poorly on downstream analysis tasks related to the ATAC modality (Fig. 4). This trade-off of sacrificing the performance of other tasks (e.g. reconstruction or prediction) to encourage better mapping between data modalities demonstrates the bias in current methods towards modeling RNA data over accurately representing ATAC. This is problematic if we hope to leverage multimodal data for ATAC-focused analyses such as trajectory inference, clustering, and differential accessibility.

## Methods

### The scPair framework
The scPair framework is outlined in Fig. 1 and Supplementary Fig. 1. It consists of two interconnected feedforward neural networks (FFN) that each accept one single cell data modality as input and predicts the other data modality as output. Each predictive network contains an encoding component to transform the input features into a low dimensional cell state space, as well as a decoding component to predict the output from the cell state.

Specifically, the last hidden layer of 30 nodes, by default, in each FFN serves as a cell state space layer that summarizes the encoded information from the input features that are most predictive of the output, and the structure between the input and cell state layer comprise the encoding network. The direct connection between the cell state layer and output provides a decoding function to transform the encoded representations into predicted output values.

The default encoding network has two hidden feedforward blocks (with 800 and 30 nodes by default) between the input and cell state layers, and each fully connected linear layer within it is followed by layer normalization[91], a LeakyReLU activation function for non-linear transformation, batch normalization[92], and 10% dropout regularization[93], sequentially.

Bidirectional networks, termed cell state mapping functions, are implemented to interconnect the cell state spaces specific to each FFN and are trained to map data from one modality to the other. By default, each of them employs a linear transformation to link modality-specific cell state spaces. These mapping functions allow integration of the two modality-specific cell state spaces by designating one as a query and the other as a reference. When available for a particular dataset, batch factors are encoded as one-hot vectors and concatenated twice, once to the input features and once to the cell state layers as additional nodes, providing both encoding and decoding networks the information about covariates.

scPair training is executed in three steps, described below:

### scPair training step 1: paired multimodal data prediction and representation learning
The initial training phase focuses on training the two predictive FFNs independently using paired single cell multimodal data, such as matched scRNA-seq and scATAC-seq profiles. Each FFN accepts one data modality as input, and predicts the other data modality as output. Appropriate loss functions have been implemented and are selected based on the data type, including the negative log-likelihood of the negative binomial (NB) distribution or zero-inflated negative binomial (ZINB) distribution for raw count output (e.g. for scRNA-seq raw counts), negative log-likelihood of the Bernoulli distribution for binary accessibility output (e.g. for scATAC-seq), and mean-squared-error (MSE) for continuous output data (e.g. for electrophysiological data, ADT data, and also for scaled RNA data). The scPair framework is implemented in PyTorch[94]. Canonical Xavier initialization[95] is applied for parameter initialization and the Adam optimizer[96] is used to tune network weights to minimize prediction loss. To control for potential confounding factors, batch labels and other covariates are incorporated into the input and cell state layers, as described above. By default, the multimodal data set

is partitioned into the training (70%), validation (10%), and held-out test (20%) sets, balanced across known biological and technical groups or batches. Early stopping after 25 tolerance epochs with no improvement in validation loss is implemented to prevent overfitting to the training data.

### scPair training step 2: paired multimodal representation (cell state) mapping
After the training of predictive FFNs, the FFN parameters are frozen and the bidirectional networks connecting the modality-specific cell states spaces are trained. The encoding networks non-linearly transform each data modality into a low-dimensional cell state space, and the bidirectional networks are optimized to directly predict a position in the cell state space of the target data modality, given a position in the source cell state space. Given the same training data used to train the predictive FFNs in step 1, we compute the 'ground truth' cell state spaces in both data modalities of the training data, then optimize the parameters of the mapping functions via MSE minimization to map between the cell state spaces.

### scPair training step 3: scPair refinement using unimodal atlases (optional)
One of the experiments conducted in this study examined whether scPair training with multimodal data could be augmented with unimodal data, in order to improve performance. In this experiment, scPair was first trained on paired multimodal data according to the procedures outlined in the two steps listed above, focused on cross-modality prediction and cell state mapping. Subsequently, the encoding networks within the pair of predictive FFNs were updated using unimodal datasets, as follows. Given the multimodal data-trained scPair model, for each unimodal training sample, we calculate the sample's position in cell state space. This yields input-output pairs consisting of the original input features and the cell state position of each unimodal training sample. We then retrain the entire corresponding encoder network with random initialization[95] in a supervised fashion using the input-output pairs derived from both the unimodal and bimodal training data, with MSE loss. Conceptually, this additional training step forces scPair to capture feature covariance from the larger and more deeply sequenced unimodal datasets, which presumably is more robust than the corresponding modalities from the multimodal dataset.

### scPair data input
scPair is flexible and currently accepts any pair of data types that can be formatted as a pair of vectors as input, representing two different data modalities measured on the same cell. For example, in experiments with paired RNA and ATAC measurements of single cells, scPair can directly utilize the raw or normalized gene expression counts for RNA-seq data, and binarized peak counts indicating accessibility status for each genomic region for ATAC-seq data. The input features can optionally be centered and scaled. By default, scPair trains its FFNs to predict one vector (corresponding to one data modality) from the other vector (corresponding to the other data modality). scPair does not require prior feature selection or filtering of the input features beyond the initial quality control (QC) steps, as the framework inherently performs (implicit) feature selection during training.

### Feature pre-selection for ATAC and RNA modalities
We evaluated two sets of scPair models: (1) without feature pre-selection, retaining the full feature sets by default, and (2) with pre-selection. We performed feature pre-selection using the Seurat::FindVariableFeatures and Signac::FindTopFeatures functions for RNA and ATAC modalities, respectively. For performing feature pre-selection for the StabMap experiments, we followed StabMap's specific feature selection

procedure outlined in its tutorial: https://marionilab.github.io/StabMap/articles/stabMap_PBMC_Multiome.html.

## Execution of generative model-based methods for single cell analysis

We followed the guidelines in the following tutorials for executing the generative variational autoencoder (VAE)-based methods for single cell multimodal analysis using their default optimal settings. Notably, to mitigate a technical issue running Polarbear, we implemented a semi-supervised framework that we termed Polarbear* that extends Polarbear[41] by using the state-of-the-art methods scVI[77] and peakVI[83] as VAEs for the RNA and ATAC modalities respectively, and implementing the same bidirectional mapping networks as scPair.

- scVI: https://docs.scvi-tools.org/en/stable/tutorials/index_scrna.html
- PeakVI: https://docs.scvi-tools.org/en/stable/tutorials/notebooks/atac/PeakVI.html
- Polarbear: https://github.com/Noble-Lab/Polarbear
- MultiVI and MultiVI*: https://docs.scvi-tools.org/en/stable/tutorials/notebooks/multimodal/MultiVI_tutorial.html
- Cobolt: https://github.com/epurdom/cobolt/blob/master/docs/tutorial.ipynb

To ensure a fair comparison, the dimensionality of the encoding hidden layers and modality-specific or joint representation layers of all benchmarked methods were configured to their optimal settings as specified in the original publications or tutorials, including their respective training procedures. For Polarbear, the bidirectional mapping networks between the embeddings of RNA VAE and ATAC VAE were initialized and trained in the same way as in scPair. Methods such as MultiVI and Cobolt apply mixture-of-experts (MoE) and product-of-experts (PoE), respectively, to model joint embeddings directly from the modality-specific representations. Thus, no explicit mapping networks are needed to bridge the modalities for these methods. The same training and held-out sets were used for all benchmarked methods. Specifically, in the benchmark experiments (Figs. 2, 3, 4), MultiVI refers to the default framework with an additional modality penalty term in the loss function to encourage the RNA and ATAC state spaces to be similar[48], while MultiVI* refers to the basic mixture-of-expert framework without penalizing the modality divergence during optimization. Parameter optimization was performed on the training set, while model performance and generalizability were evaluated on the held-out set.

## Performance evaluation overview

We evaluated the performance of the benchmarked methods on the tasks of cross-modality prediction and cell state mapping. In our setup, the held-out test sets for both RNA and ATAC data were excluded from model training process. During testing, we alternated between providing one data modality of the held-out data as input, and using the other data modality as ground truth to evaluate predictions. For example, the ATAC peak profile of a held-out cell would be provided as input to the trained model, and the predicted RNA profile would be evaluated against the ground truth RNA expression measurement to quantify prediction accuracy.

We measured performance using several measures described below; first, we will introduce our notation. Let $\mathbf{x}_n$ and $\mathbf{y}_n$ denote the measured gene expression and chromatin accessibility profiles respectively for held-out sample $n$, and their corresponding predicted profiles are denoted as $\hat{\mathbf{x}}_n$ and $\hat{\mathbf{y}}_n$. $\mathbf{p}_n$ and $\mathbf{q}_n$ represent the cell states estimated by the scPair model based on $\mathbf{x}_n$ and $\mathbf{y}_n$, respectively (e.g. the RNA encoding network of scPair accepts $\mathbf{x}_n$ as input and estimates the RNA-specific cell state $\mathbf{p}_n$). Furthermore, let

$\hat{\mathbf{p}}_n$ represent the mapped RNA cell state from ATAC-specific cell state $\mathbf{q}_n$, and $\hat{\mathbf{q}}_n$ represent the mapped ATAC cell state from the RNA-specific cell state $\mathbf{p}_n$ via the bidirectional networks. Finally, let $T$ denote the indices of samples belonging to the held-out test set. To quantify the performance of each benchmarked method, we applied established metrics for different tasks[97], defined below.

## Cross-modality cell state mapping accuracy

To quantify the accuracy of mapping cell states across modalities, we calculated the pairwise Euclidean distance between each predicted cell state (based on the other data modality) and actual cell state. In the ideal scenario, given a collection of samples from the held-out test set, the nearest neighbor to the true cell state $\mathbf{p}_n$ (based on RNA) should be the predicted cell state $\hat{\mathbf{p}}_n$ (based on ATAC, the other modality) for the same cell. The predicted cell states for other cells $\hat{\mathbf{p}}_m$ should be farther in distance from $\mathbf{p}_n$ compared to true match $\hat{\mathbf{p}}_n$. We therefore use the Fraction Of Samples Closer Than the True Match (FOSCTTM) metric to evaluate the cell state mapping accuracy[98]. Lower FOSCTTM indicates better mapping performance. For example, to evaluate ATAC→RNA mapping accuracy:

$$\text{FOSCTTM} = \frac{1}{|T|} \sum_{n \in T} \left( \frac{1}{|T|-1} \sum_{m \in T, m \neq n}^{T} \left[ ||\hat{\mathbf{P}}_m - \mathbf{P}_n||_2^2 < ||\hat{\mathbf{P}}_n - \mathbf{P}_n||_2^2 \right] \right)$$

(1)

## RNA expression prediction accuracy

To evaluate the accuracy of predicting RNA expression profiles from chromatin accessibility data, we calculated the Pearson correlation coefficient between the ground truth (measured) gene expression profile and predicted gene expression profile from chromatin accessibility for each held-out test cell:

$$\text{PCC} = \frac{1}{|T|} \sum_{n \in T} \text{Pearson}\left( \hat{X}_n, X_n \right)$$

(2)

Higher correlation values on the held-out test set indicates more accurate prediction.

## Chromatin accessibility prediction accuracy

To evaluate the accuracy of predicting binary chromatin accessibility profiles from gene expression data, we calculated the area under the ROC curve (auROC) between the binarized ground truth chromatin accessibility profile and the predicted profile for each held-out test cell. An auROC of 1 indicates perfect prediction, while 0.5 represents the performance of random predictions.

## Cell population clustering accuracy

We evaluated mapping methods in part based on how well cells of the same cell type clustered in cell state space. To do so, we calculated the Adjusted Rand Index (ARI). The ARI metric compares two different clustering assignments: (1) the model-based clusters derived from the low-dimensional cell state spaces learned by each method, and (2) the clusters based on the pre-defined cell type labels. An ARI value of 1 indicates a 1–1 matching between the clusters identified by the model and the ground truth cell types, while an ARI value of 0 indicates no coherence between the model's clusters and the ground truth cell types. Let $a_i$ ($i = 1, 2, ..., K$) denote the number of cells in the $i$ th cluster from the ground truth cell type labels and $b_j$ ($j = 1, 2, ..., K$) denote the number of cells in the $j$ th cluster from the model. Let $n_{i,j}$ denote the number of observations in both the $i$ th cluster from the ground truth cell type labels and the $j$ th cluster from the model. The ARI is

calculated by:

$$ARI = \frac{\sum_{ij}\binom{n_{i,j}}{2} - \left[\sum_i\binom{a_i}{2} + \sum_j\binom{b_j}{2}\right]/\binom{n}{2}}{\frac{1}{2}\left[\sum_i\binom{a_i}{2} + \sum_j\binom{b_j}{2}\right] - \left[\sum_i\binom{a_i}{2} + \sum_j\binom{b_j}{2}\right]/\binom{n}{2}} \quad (3)$$

Specifically, the ARI calculates the similarity between the two assignments while accounting for chance grouping, where larger values indicate better performance. It quantifies the percent overlap between the model-based clusters and known cell types, adjusted for the expected overlap by chance.

### Pseudotime inference consistency between modalities
In a number of our experiments on bimodal datasets, we evaluated the consistency of pseudotime inference performed on the RNA and ATAC modalities separately. Here we use the Spearman correlation between RNA- and ATAC-inferred pseudotimes for the same cells to measure consistency.

### Electrophysiological feature prediction accuracy
To evaluate the accuracy of predicting the standardized electrophysiological features from Patch-seq transcriptomic data, we calculated both the Pearson correlation coefficient and mean-squared-error (MSE) between the predicted and ground truth electrophysiological measurements.

### Visualization of modality-specific cell state spaces
To visualize the learned cell state spaces for each data modality, we extract the output of the last layer of the encoding networks (cell state space layer) within the modality-specific predictive FFNs. To visualize these cell state spaces, we apply Uniform Manifold Approximation and Projection[27] (UMAP) on the cell state vectors to enable visualization in two dimensions via the umap-learn (v0.5.5) Python package.

### Dataset preprocessing overview
The datasets and preprocessing steps utilized in this study are described below. Notably, assays such as 10x Genomics Chromium Single Cell Multiome ATAC + Gene Expression (10x scMultiome) collect measurements from nuclei, but since the official name uses the term "single cell", we use the term single cell (and not single nucleus) to maintain the original naming convention. scPair, as well as other approaches tested here, are similarly applied to single cell versus single nucleus datasets. We applied feature-wise quality control (QC) to each one of the following datasets during processing by removing features (genes for RNA, peaks for ATAC) that were detected in <1% of the total population to eliminate technical noise. Below, we list additional QC steps performed per study.

### sci-CAR cell line dataset processing
Cao et al.[55] used the sci-CAR assay to jointly profile RNA and chromatin accessibility (ATAC) in cells from the 3T3, 293 T and A549 cell lines. This dataset consists of 4,216 cells, with 17,368 genes and 57,425 peaks post filtering from the original study that have been used in this analysis.

### 10x scMultiome human peripheral blood mononuclear cells (PBMC) dataset processing
The 10x Genomics Single Cell Multiome peripheral blood mononuclear cell (PBMC) dataset (pbmc_granulocyte_sorted_10k) contains joint profiling of gene expression (RNA-seq) and chromatin accessibility (ATAC-seq) in individual cells. The raw count matrices consist of 11,909 human PBMCs. Quality control was performed following the workflow outlined in the Signac[50] software package tutorial

(v1.10.0, https://stuartlab.org/signac/) in R, and cells were filtered based on RNA and ATAC quality metrics, including the number of RNA counts (1,000-25,000), number of ATAC counts (1,000-100,000), nucleosome signal (<2) and TSS enrichment ( > 1). 11,331 high-quality cells remained after QC. Specifically, for the ATAC modality, peaks were re-called using the built-in MACS2-based[99] CallPeaks function. The final curated dataset contained expression profiles for 13,515 genes and accessibility profiles for 86,002 peaks across the 11,331 cells.

### 10x scMultiome mouse brain dataset processing
The 10x Genomics Single Cell Multiome mouse brain data[56] contains joint profiling of gene expression (RNA-seq) and chromatin accessibility (ATAC-seq) across 9385 individual cells. After excluding a less representative cell type (cluster 0) consisting of 15 cells, the dataset was reduced to 9370 cells. Following preprocessing, the dataset retained 14,461 genes and 82,474 peaks.

### SHARE-seq mouse skin dataset processing
The SHARE-seq dataset from Ma et al.[32] contains joint profiling of RNA expression and chromatin accessibility for 34,774 cells from mouse skin tissue. 15,436 genes and 97,669 peaks were retained after preprocessing. We generated the corresponding tabix index file for the fragments file as suggested by the authors.

### SNARE-seq mouse adult cerebral cortex dataset processing
The SNARE-seq mouse adult cerebral cortex dataset from Chen et al.[33] contains joint transcriptional and epigenomic profiling of single nuclei from the adult mouse cerebral cortex. Based on RNA and ATAC quality metrics, including the number of RNA counts ( > 800), number of ATAC counts (>500), blacklist fraction (<0.03) and TSS enrichment (<20), 8055 high-quality nuclei remained after QC. The final dataset contained expression profiles for 9104 genes and accessibility profiles for 37,030 peaks. Metadata was provided directly by the original study, and the fragments file was downloaded from the Signac tutorial (https://stuartlab.org/signac/articles/snareseq).

### SNARE-seq2 primate primary motor cortex (M1) dataset processing
Bakken et al.[54] utilized the SNARE-seq2 technique to jointly profile the RNA expression and chromatin accessibility of nuclei isolated from the primary motor cortex (M1) of human and marmoset. The authors provided preprocessed cell-by-feature count matrices and sample metadata of 84,178 human nuclei and 9,946 marmoset nuclei that were used for analysis in our study. Feature QC followed the same criteria as described above, resulting in 16,139 genes and 33,517 peaks in human, and 20,292 genes and 35,573 peaks for marmoset.

### SNARE-seq mouse neonatal cerebral cortex dataset processing
Chen et al.[33] applied the SNARE-seq assay to jointly profile the RNA expression and chromatin accessibility of 1,469 single nuclei derived from cortical tissue of newborn (P0) mice. We focused our analysis on five cell populations: 214 IP–Hmgn2, 99 IP–Gadd45g, 437 IP–Eomes, 177 Ex-L2/3–Cntn2, and 542 Ex-L2/3–Cux1. 12,047 genes and 120,838 peaks were retained after preprocessing.

### Unimodal scRNA-seq and snATAC-seq mouse cortical dataset processing
The mouse isocortical transcriptomic single-modality data (scRNA-seq) was acquired from the Allen Brain Atlas[46,100] with 15,413 cells, and the mouse secondary motor cortex epigenomic single-modality data (snATAC-seq) was acquired from the snapATAC study[13] with 28,490 nuclei. The feature set of each modality was set to match the SNARE-seq mouse adult cerebral cortex dataset, with 9710 genes for unimodal scRNA-seq data and 58,016 peaks for unimodal snATAC-seq data.

## Allaway et al., E13 mouse scRNA-seq, scATAC-seq, and scMultiome dataset processing

Single cell paired and unpaired multiomic datasets from the E13 mouse medial ganglionic eminence (MGE) region were obtained[65]. After cell QC steps based on quantity metrics including the number of RNA counts (300–30,000), the number of RNA features (200–6000), the number of ATAC counts (300–60,000), the number of ATAC features (300–30,000), percentage of ribosomal genes (2%–30%), nucleosome signal (<1.5) and TSS enrichment (3–15), performed using Seurat (v4.3.0) and Signac (v1.10.0) in R, the dataset contained 11,258 cells in the unimodal scRNA-seq dataset, 24,589 cells in the unimodal scATAC-seq dataset, and 5,308 cells with paired multiomic profiles. For the analysis, we adopted the strategy used by the authors of the original paper, focusing exclusively on postmitotic cells. This led to a final count of 14,605 unimodal scATAC cells and 2,141 multiomic cells. The number of common genes and peaks were 14,450 and 142,402, respectively.

## Patch-seq mouse GABAergic neuronal dataset processing

Gouwens et al.[35] collected Patch-seq profiles consisting of RNA paired with electrophysiological recordings for GABAergic neurons from the mouse primary visual cortex. We obtained the single cell RNA SMART-seq data from the Allen Brain Map Portal (https://portal.brain-map.org/cell-types/classes/multimodal-characterization), and the electrophysiological (ephys) data was downloaded from the Distributed Archives for Neurophysiology Data Integration (DANDI, ID:000020). We processed the transcriptomic data using Seurat (v4.3.0) in R and excluded the Meis2 cell type, as it has been reported as a distant branch preceding the major glutamatergic and GABAergic split[100]. We also removed the aspiny dendrite type due to their substantial difference in transcriptomics profiles from the remaining GABAergic cell population[15]. For the ephys data, we performed feature extraction on raw recording data stored in NWB files using the AllenSDK (v0.16.3) and IPFX packages (v1.0.4) in R. Following discussions with the author of the original study[35], we further processed the ephys data and extracted 11 feature vectors, computed their principal components[15,35] and finally obtained 3,302 GABAergic neurons passing QC on both modalities, with 29,197 genes and 35 IPFX features plus 76 PCs for 11 feature vectors. In the ephys modality, extreme outlier values for each ephys feature were defined as values beyond the 0.01 and 0.99 percentiles, and removed by being marked as NA. The processed data was then standardized by centering and scaling, ignoring the NAs. Values more extreme than six standard deviations was set as 6 to be consistent with the related studies from Allen Brain Institute[35,101].

## Unimodal mouse GABAergic neuronal scRNA-seq dataset processing

We obtained a dataset of 22,521 GABAergic neurons sequenced using SMART-seq to act as a unimodal transcriptomic dataset for augmenting scPair training on the Patch-seq mouse GABAergic neuronal dataset[46]. We intersected the gene features to match the Patch-seq transcriptomic data at 29,197 genes.

## 10x human peripheral blood mononuclear cells (PBMC) CITE-seq dataset processing

The CITE-seq PBMC data (pbmc_10k_protein_v3) was provided from 10x Genomics and was downloaded via the scvi.data.dataset_10x() function. It contains joint profiling of gene expression (RNA) and Antibody-Derived Tags (ADT) profiles in individual cells. The preprocessed matrices consist of 6855 human PBMCs with 15,677 RNA features and 14 ADT features.

## Genome browser track visualization

The visualizations of chromatin accessibility data as genome tracks were performed using the *CoveragePlot* function from Signac (v1.10.0) package in R, based on the fragments bed files of the ATAC-seq datasets.

## Assignment of postmitotic state for bimodal MGE data

The E13 MGE data from Allaway et al.[65] consisted of unimodal and multimodal single cell data. In this dataset, the mitotic states (mitotic or postmitotic) of the unimodal scRNA-seq and scATAC-seq cells were established from fluorescence-activated cell sorting (FACS) during data generation[65], but was unknown for the multimodal single cell data. In the original paper, the authors restricted their analysis to postmitotic cells. We therefore needed to infer the mitotic status of the multimodal cells. To do so, we first trained scPair on the full multimodal E13 mouse MGE dataset, which contained a heterogeneous mixture of both mitotic and postmitotic cells. We then computed the cell state positions of each unimodal cell using the trained scPair model. Finally, we predicted the mitotic state of the multimodal cells based on the nearest neighbor matching ($k = 1$) of each multimodal cell (unlabeled) to the unimodal cells (labeled).

## Pseudotime trajectory inference

Pseudotime trajectories were inferred for all datasets by inputting the learned low-dimensional cell states/representations by various methods, such as scPair, into the Palantir[62] (v1.2.0) Python package. Palantir was used to run a diffusion map using the palantir.utils.run_diffusion_maps() function. The resulting diffusion components were plotted to identify the root cell for each analysis based on the target component values. Afterwards, the pseudotime values and branch probabilities for each cell were calculated using the palantir.core.run_palantir() function with parameters set to num_waypoints = 1000 and knn = 30, as described in the previous study[65]. Cells were assigned to the trunk and branches based on the Palantir result, where the intermediate transient states can also be determined according to the clusters alongside the trunk and branches. Stage-specific features are then captured through differential analysis.

## Feature-wise pseudotime estimation

In order to quantify the specific pseudotime along the inferred trajectories at which a given gene or chromatin peak shows highest levels of expression or accessibility, we compute a weighted mean pseudotime $\tau_f$ for each feature following the approach of Trevino et al[102]. More specifically, the aggregated gene or peak feature counts of each feature $f$ in pseudobulk sample $i$ is denoted as $C_{i,f}$. Further, let $t_i$ represent the average inferred pseudotime across all single cells comprising pseudobulk sample $i$. The weighted mean pseudotime or "feature pseudotime" $\tau_f$ for $f$ is then calculated as:

$$\tau_f = \sum_{i=1}^{N} t_i \frac{C_{i,f}}{\sum_{k=1}^{N} C_{k,f}} \tag{4}$$

where N is the number of pseudobulk samples. Pseudobulk samples were first constructed by grouping single cells along their inferred pseudotime order such that each pseudobulk sample comprised the same number of single cells ($N = 20$ for 10x scMultiome data, and $N = 80$ for unimodal data). The pseudotime label for each pseudobulk sample was then assigned by averaging the pseudotimes of its constituent single cells. For downstream visualization purposes, rare (10th percentile) and highly abundant (90 percentile) features such as peaks and genes were removed to enable smoother data visualization.

## Pseudotime-binned pseudobulk profile generation during trajectory inference analysis

Unlike the pseudobulk aggregation strategy described above that was specific for feature-wise pseudotime estimation, a new set of pseudobulk profiles were generated from the single cell transcriptomic or chromatin accessibility data along the inferred trajectories to enable

heatmap visualization with evenly spaced pseudotime intervals (Fig. 5). To achieve this, the entire pseudotime range estimated by Palantir (0–1) was segmented into equal-width bins of 0.05 pseudotime units. Within each bin, all sample (cell) profiles were averaged to construct pseudobulk profiles. This approach ensured that each pseudobulk contained information from all measured cells, facilitating smooth variations in gene expression or chromatin accessibility patterns across the entire pseudotime spectrum.

## Motif enrichment analysis

Transcription factor binding motif enrichment analysis was performed by leveraging the position frequency matrices from the JASPAR[103] database (v2022) in R, together with the ChromVAR[67] (v1.20.2) and Signac[50] (v1.10.0) R packages for incorporation of motif information into the single cell chromatin accessibility data object. The RunChromVAR() function was applied to perform motif enrichment analysis, which identifies regulatory motifs exhibiting significant variability across cells (adjusted $p$-value < 0.05 based on Bonferroni correction and log2 fold change > 1). To visualize results, the MotifPlot() function was utilized to generate motif sequence logos showing information content for each motif position.

## Incorporation of large unimodal datasets using scPair and StabMap

Initially, both scPair and StabMap were trained on the same multimodal SNARE-seq mouse adult cerebral cortex dataset with a predefined held-out test set. Subsequently, separate training sessions incorporated unimodal scRNA-seq and scATAC-seq mouse cortical datasets into each method. Finally, both unimodal datasets were jointly incorporated into each method's training process. Mapping accuracy was then calculated across four training scenarios: (1) without incorporating unimodal data, (2) incorporating only scATAC-seq data, (3) incorporating only scRNA-seq data, and (4) incorporating both unimodal scATAC-seq and scRNA-seq data, based on the held-out test set. Notably, StabMap requires feature selection and log-normalization of the count matrices. Therefore, we followed the tutorial and applied the functions modelGeneVar() and logNormCounts(), where for the RNA and ATAC modalities, the filtration mean threshold is 0.01 and 0.05 respectively, and the p-value threshold is 0.05.

## Statistics and reproducibility

All statistical calculations were implemented in Python (v3.10; https://www.python.org) or R (v4.2.2; https://cran.r-project.org). The detailed statistical tests are indicated in corresponding figure legends/captions where applicable. No statistical method was used to predetermine sample size. The experiments were not randomized. This study does not involve group allocation that requires blinding.

## Reporting summary

Further information on research design is available in the Nature Portfolio Reporting Summary linked to this article.

## Data availability

The following publicly available datasets were analyzed in this study: sci-CAR cell line dataset (NCBI GEO: GSE117089), 10x scMultiome PBMCs dataset (10x Genomics), 10x scMultiome mouse brain dataset (NCBI GEO: GSE184981), mouse skin SHARE-seq data (NCBI GEO: GSE140203), P0 mouse and adult mouse cortex SNARE-seq data (NCBI GEO: GSE126074), marmoset and human cortex SNARE-seq2 datasets (Neuroscience Multi-omics Archive), mouse cortex unimodal scRNA atlas (Allen Brain Cell Atlas-mouse whole-brain cell-type atlas), mouse cortex unimodal scATAC atlas (NCBI GEO: GSE126724), mouse GABAergic neuron Patch-seq data (transcriptomic data); the DANDI Archive raw electrophysiology data: 000020, E13 mouse MGE

unimodal and multimodal data (NCBI GEO: GSE165233), and CITE-seq PBMC data (pbmc_10k_protein_v3 from 10x Genomics and downloaded via scvi.data.dataset_10x function). No new sequencing data was generated. The processed benchmarking datasets can be accessed at https://figshare.com/s/ea98335f6f8a0abc8ae5. All other data supporting the findings of this study are available within the article and its supplementary files. Any additional requests for information can be directed to, and will be fulfilled by, the lead contact. Source data are provided with this paper as a separate **SourceData** file.

## Code availability

Source code[104] and tutorial are available at https://github.com/quontitative-biology/scPair. The version of scPair used for the analyses presented in this paper has been deposited to Zenodo: https://zenodo.org/records/12735193.

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

## Acknowledgements

This work was supported by NSF CAREER award (1846559 to G.Q.). This has been made possible in part by grants from the National Institutes of Health (NIH), including the Office of the Director/National Institute of Mental Health (DP2 MH129987, G.Q.), and the National Institute of Child Health and Human Development (P50 HD103526).

## Author contributions

G.Q. conceived and supervised the project. G.Q. and H.H. designed the scPair framework and benchmark evaluations, and wrote the manuscript. H.H. implemented the prototype of scPair framework, conducted the benchmark and validation analyses. Cartoons and illustrations in Fig. 1 were created with BioRender.com and Adobe Illustrator by H.H.

## Competing interests

The authors declare no competing interests.
