## [Transparent Peer Review file · Nature Communications]

scPair: boosting single cell multimodal analysis by leveraging implicit feature selection and single cell atlases

Corresponding Author: Professor Gerald Quon

Version 0:

Reviewer comments:

Reviewer #1

(Remarks to the Author)

Highlights:

The current analysis approaches for single-cell multimodal data face two challenges: 1) $p \gg n$ problem, where the total feature space (p), such as that combining gene expression, chromatin openness, etc., are hugely more than the number of cells loaded (n); and 2) data quality, where the relatively shallow sequencing depth might result in unstable/unreliable measurements of features. The authors, Hu and Quon, developed a deep learning framework called scPair that mitigates these issues and outperforms other tools in terms of better cell state inference. Generally speaking, scPair works on single-cell assay that measures two modalities simultaneously, training two encoders, each of which projects one modality to the other, and a third NN that bidirectionally translates between the two encoders. One excitement of Hu and Quon's approach lies in its outstanding performance of distinctive cell-state clustering in the scATAC modality, which used to be a challenging area. The other highlight of scPair is that it can optionally borrow information from unimodal measurements to further enhance the performance, overcoming the shallow sequencing weakness of multi-modal data. Using scPair, Hu and Quon identified transcription factors (TFs) that are specific to different neural differentiation stages.

Major Comments:

1. Can authors quantitatively comment on the underlying factors that lead to the performance gain in scPair by setting up some head-to-head comparisons? Is that because scPair uses all features without pre-selection before inputting the model, or is that due to the cross-modality structure of the scPair model? The former should be easy to quantify by training two scPair models with and without feature selection before feeding and comparing the performance difference. This will be a critical insight in the light of shaping a more robust preprocessing workflow for the overall single-cell analysis, especially for scATAC. The latter question about the model structure advantage can be achieved by comparing scPair to a more similar model, for example, scMDC (PMID: 36513636) that was also published in Nat Comm. scMDC's model structure also includes the ability to project one modality to the other and has an encoder layer that implicitly selects the most important features for the two modalities, making it more comparable to scPair. On the other hand, scMDC concatenates the two modalities before the encoder layer so that the encoder layer is a mixture. This key difference will help us understand the benefit of training two FNNs and a connecting NN (e.g. scPair) over training a single GAN (e.g. scMDC).
2. It is unclear how the authors implemented "the dimensionality of the encoding hidden layers and modality-specific or joint representation layers of all benchmarked methods were set to match the cell state dimensions in scPair" (Line 562-564). Is it done at the model-setup stage before training or at the post-training stage, where the encoder space is truncated to fit the same size as that in scPair? In fact, different models' latent spaces may have learned different "biological" contexts. For example, the RNA encoder layer of scPair learns the latent features that are most representative of ATAC, but in Polarbear it learns the latent features that are most representative of gene expression. Therefore, forcing the same dimensionality as scPair won't make it fair for other methods. If the authors kept other tools' optimized dimensionality in the latent space and only re-compared the performance in all the matrices, would the conclusion change systematically (e.g. Figures 2c-d and 3a-b)? This should be at least included as a supplementary.
3. I had a bit of a hard time getting my head around: both Fig. 2a and Fig. 3a compare the performance of RNA-to-ATAC projects across tools on different datasets. While I understand the two figures using two different matrices, namely (1-FOSCTTM) and Pearson correlation, why the overall trends are quite consistent on the four SNARE-seq datasets but different on 10X? Especially, why Polarbear is close to scPair on 10X in Fig. 2a but suddenly far away from scPair on 10X in Fig. 3a? Theoretically, (1-FOSCTTM) and Pearson correlation should be correlated/translatable?
4. I'd encourage the authors to improve the text for Figure 3e-g so that the general audience who are not familiar with the

benchmark datasets can easily recognize the power and robustness of scPair. Specifically, what are the tissues and expected cell types/clusters? What are the known gene markers of each type/cluster and how do the four ATAC regional examples selected by the authors support those expected gene markers? The authors' own description of neonatal mouse brain SNARE-seq dataset is a good exemplar to follow.

5. In Fig. 4e, it is impressive to see the strong alignment of key gene expression between the scPair pseudotime and the real-world development time. However, I have some questions about the selection of these key genes: 1) first of all, I am wondering why not all genes are colored according to the development stages, which are supposed to be purple, red, green, blue and orange according to other Fig. 4 panels. What do the gene names in black mean? 2) the authors claimed these genes are adapted from Chen et al.'s Nat Biotech paper in 2019. But some genes in Fig. 4e are NOT the markers of the five stages in the original studies. For example, Vim was reported by the original authors as a marker for Ast/RG, not any of the five stages. And Notch1 was NOT in the original marker list. These discrepancies might be due to the fact that the author re-analyzed the original data by subsetting to five stages instead of using the whole data. If the authors used the markers of the five stages from the original study's Supplementary Table 1, which I think is a more robust estimation with more cell types involved, what would the new Fig. 4e look like compared to MultiVI's outcome?

6. While the authors described a complex process of their approach of integrating unimodal scATAC data into their scPair models to improve the performance from Line 250-255. But the language there needs to be corrected and orchestrated in a better way, as this is the most critical technical part that will help the readers to digest the rest of the paper. Currently, it reads not only repetitive/circulated but also broken. For example, 1) "We first perform trajectory inference on the RNA component... more successful in RNA space" which is fine, but the next sentence, "The RNA cell states used as input to trajectory inference are...", talks about the same thing again and seems to be unnecessary. 2) "Next, we predict the RNA cell ..." is fine. But the next immediate part, "use the trained scPair model to infer the cell state of each cell in the unimodal scATAC dataset" comes from nowhere. Is this a part going to be removed? In addition, I'd also suggest the authors include a main figure here to facilitate the demonstration. (This can be Fig. S1d if taking Q13 below into account is feasible.)

7. The number of cells involved in the scPair's implementation on the Allaway et al. dataset is quite confusing and needs to be better clarified. Line 268 says there are 24,589 unimodal scATAC cells. But Line 280 says the pseudotime assignment is only to 14,605 unimodal scATAC cells. Is this because the missing (~10,000) unimodal scATAC cells have been consumed in the training process? It sounds contradictory to Line 258 where the authors claim that "each unimodal scATAC-seq cell will have an assigned pseudotime".

8. What is the downsampling strategy used in "Pseudotime-binned pseudobulk profile" toward the bin with the lowest counts? Can the authors comment on whether the cells with important features might be filtered out during this process? In other words, can the authors justify their downsampling strategy would lead to reproducible results in Fig. 5d?

9. Fig. 5e bottom panel, does the x-axis actually represent the pseudo-time of "RNA cell states predicted from" Unimodal scATAC? If so, it should be labeled precisely. The current label of "Unimodal scATAC-seq pseudotime" is misleading and has no difference from that in the bottom panel of Fig. 6d. Similarly, in Fig. S7d, the x-axis seems better to be "Pseudo time based on RNA cell states predicted from 10x scMultiome ATAC".

10. Line 309, the author writes "The improved identification of branch-specific genes is not driven by our use of scPair". I might get where the authors are trying to reach with this sentence, but I don't agree with the current way of expressing it. In contrast, the improvement is indeed driven by the scPair by leveraging the unimodal data. The failure to display the off-diagonal trends in the heatmaps in Fig. S7d might indicate that the training of scMultiome ATAC RNA projection was not successful due to the issue of data quality/low cell number; yet scPair can help to overcome this caveat by borrowing information from unimodal scATAC. This is only my hypothesis, and the authors still justify whether it is correct.

11. Fig. 5g provides highly valuable insights into neurodevelopmental regulation. To enhance the excitement of scPair application and the biological impact of these findings, can the author also provide a list of genes associated with each set of chromatin openness? Specifically, if each set of chromatin openness is forced to be zero in the ATAC RNA FNN, which set of genes is most influenced on the output side?

12. The gain from scPair-augment on Patch-seq is quite modest. This seems to suggest that even if the sequencing depth in the Patch-seq is low, they still robustly measure the key features that largely contribute to dimension reduction. While this is not to say scPair-augment is totally useless, the authors may want to retain Fig. 6c-d in the supplementary figures.

13. A final thought and not necessarily to be addressed: given the inclusion of unimodal scATAC indeed leads to an improved biological interpretation on the Allaway et al. dataset, I am wondering whether the authors can 1) absorb scPair-augment implementation on the Allaway et al. dataset and replace the scPair-standard there, and 2) land on all the same conclusions as in the current Fig. 5. If so, that can be an ideal place to establish the power of scPair-augment, against low cell number instead of against low sequencing depth, with critical and novel biological insights.

Minor Comments:

1. The author should provide better guidance on installing the scPair dependencies. Specifically, I found that three packages, namely "intel-openmp", "cobolt" and "jaxlib", couldn't be installed directly from the provided yml file. I had to install "cobolt" and "jaxlib" manually. I still couldn't install "intel-openmp" and am wondering if it is necessary.

2. The authors interchangeably wrote Polarbear, PairedVAE, and Polbear/pairedVAE in the text and figure legends. Please use a consistent nomenclature.

3. Is there a reason Cobolt's performance is not included in Figure 3b?

4. The authors repeatedly cite the same methodology and data source papers, which I think is unnecessary. To name a few, check the occurrence for references 33, 59, and 60. I'd suggest citing them once they are first mentioned in the main text. Then keep using the same term throughout the paper without citing them again.

5. Figure S6, the author names are used on the plot but the database names are used in the legend. It is better to unify them for easy reading and understanding.

6. Line 282, "three terminal terminals"?

7. Fig. S7b and S7e-f are not mentioned in the main text. I believe Fig. S7b should have been cited along with Fig. 5b, c, and Fig. S7c in Line 289.

8. Throughout the section “Unimodal datasets increase the density of cells along multimodal trajectories”, the word “scATAC-seq” sometimes is used alone without the attributive “unimodal” in front, which might cause confusion.
9. Fig. 6b is better plotted according to (1-FOSCTTM) to visually display gains from using unimodal data. This way is also consistent with Fig. 2c.
10. Due to the amount of datasets used in this paper, can authors provide a supplementary table that associates the analyses/benchmarks with their involved datasets?

Reviewer #2

(Remarks to the Author)

The idea to start from all features to feed a FFN instead of feature selection, as selecting the n hypervariable features could lead to bias, looks new and valid to me. Although, there might be some overfitting as well (see comments). The metric to perform the benchmarking to other tools are well-chosen, but only the standard mode is benchmarked. The Python implementation could be more user-friendly and documentation should be more elaborate. Doing so will eventually lead to more users.

Major Comments

- The authors state that “we intuitively expect that there should exist a mapping between RNA-based cell state and ATAC-based cell state”. According to my opinion chromatin changes occur before induction of transcriptional changes in time. Is the ATAC-based cell state than not a projection of the current and future RNA-based cell state? Will scPAIR takes this into account and if not, does scPAIR is not overcorrecting?
- In the introduction there something mentioned about the higher depth of unimodal data. This is chosen by the data generator and is not intrinsic to the methods themselves as you can define yourself how intensive you will sequence every pool.
- For most datasets there is a big difference in the mapping accuracy and the prediction accuracy between the PBMC dataset and others. What is the reason for that? Is this intrinsic to the method? 10Xgenomics vs all others? Or is either sequencing depth related? A second 10X genomics dataset would be of value to see if it is related to the methodology or the sample type.
- How scalable is scPAIR (both modes) to large datasets (100k->1Mcells)?
- The number of cells and reads per cell clearly have an impact on the integration (see trajectory part) thus what is the minimum number of cells (for the “bridging population”), read coverage needed to perform good enough?
- scPair augment is not benchmarked against methods that work with a bridge population. Is this strategy better performing compared to similar methods?
- In the discussion the authors suggest that the method might work on other multimodal datatypes Eg. CITE-seq data, nanoCUT&TAG data, ... I think this should be proven or otherwise it should be formulated less strong and possible struggles to use it for other methods should be mentioned more in the discussion (eg, adaptation of distributions, ...).
- The documentation on how to perform scPAIR is not complete yet. The example code should be better documented. Ideally scPAIR starts from a multiomics anndata object and is performed in only some lines of code (eg, only splitting data, training, testing). Now it looks like the user has to still code quite a lot.

Minor Comments

- Figure 1c: towards instead of towards
- Figure 3: put y-axis from 0-1 on both A and B to make this comparable
- From the introduction and abstract it should be more clear which type of multimodal data is needed. By reading the text bot paired and also unpaired with bridge populations is possible.
- The introduction part is not always clear/to the point.
- Line 42: “and most other common tasks” is vague,
- line 53-55: about clustering unimodal data and defining parameters, this is also the case for multimodal data and is not changed after integration.
- Line 68-69: Sentence is not clear to me. You nowadays have also diagonal integration methods. Some of them use indeed bridge data, but others don't.
- Line 173-177, instead of mentioning the increase in Pearson correlation and auROC, it would be better to mention the (average) values. It's a difference when it goes low to mid range value of mid range to high, which you can not conclude now from the text.
- In the method section I would like to see additional information for the used datasets for benchmarking. How many reads (RNA/ATAC) per cell for each modality, some QC parameters (median/mean mtRNA%, TSS overlapping reads, ...) for both modalities, ..

Reviewer #3

(Remarks to the Author)

The authors apply neural networks to the problem of combining multi-modal, most ATAC and RNA-seq assays. This is not fully my domain, but an important problem. I mostly understand the paper, but don't know the literature in this domain, so I would not know if they have sufficiently cited the literature. It's an interesting paper, looks like it advances the field. I mostly have questions on details, no major concerns about the main results. I do have concerns about the source code which I am not able to run in its current form.

Questions:

- p 6, line 142: why are you using the FOSTTM score instead of a the usual Sensitivity/Specificity or ROC or Accuracy/PPV? I may be wrong, but I thought this was a simple class prediction problem, where the classifier predicts a label.
- p 12 | 323: You write that the motif NKX2-1 is specific for a certain type of non-coding regions. the motif of NKX2-1 is CACTTA (see UniProbe), but as far as I know, most NK-class homeodomain recognize this motif and similar ones. I am surprised that you can derive a hypothesis like this from just motif matches. Could you check is NKX2-1 is really the only predicted binding factor? I imagine there are a lot of other motifs that come up and you picked this one?

Discussion:

- I'm missing an explanation or hypothesis why ATAC seq data is mapping poorly to cell states. You have a hypothesis, could you add to the discussion?

Code and data sharing:

- code needs documentation on how to load the input data and at least one example input file and output file.
- I think you can increase citations quite a bit if you can provide the dataset. Anyone coming after you and who writes similar algorithms will need to do the same evaluation again. If you can provide the input matrices and the output prediction values for your and the other software packages will be able to use your files instead of spending a lot of time (x weeks?) on finding the input data from the other papers. You could upload them to figshare as a zipfile.
- For example, I tried to find your input data on Nemo, you helpfully provide the link <https://assets.nemoarchive.org/dat-ek5dbmu> but then I don't know which of the hundreds of files you used for your analysis. If you had a figshare link, I could just download that.

typos:

- page 3, line 55: explicit -> explicitly
- page 5, line 09: it is derived based - not sure what is wrong here. Missing word?
- p.9 |254: missing word before the dot. not sure what it is. "We" ?
- p10 |274: we aimed to avoid -> we aimed at avoiding or rather: our aim here was avoiding
- p10 |284: we also illustrate *that*

Reviewer: Max Haeussler, Cell Browser author, cells.ucsc.edu, UCSC

Version 1:

Reviewer comments:

Reviewer #1

(Remarks to the Author)

The authors have addressed all my comments nicely. I believe the paper is ready to be accepted by Nature Communications.

Reviewer #2

(Remarks to the Author)

The quality of the paper improved greatly. I am pleased that the authors added another type of multimodal data and included more multimodal datasets in the benchmark. The documentation on their GitHub is more extensive, follow-up with user comments will be key in the future. I only have a minor comment remaining.

- Fig 3A: For the score and ranking, it would be helpful to have some reference numbers and best/worse indication.

Reviewer #3

(Remarks to the Author)

The changes look good. Thanks for your detailed replies!

One note: Reviewer 1 had the great idea of asking for a comparison with scMDC, which you did and you found that the method is too different to be compared. However, could you still mention this fact and/or attach the results as a supplemental file? I guess that many readers will think that scMDC is very similar and if they see that you already covered this case, this would help them. It would be too bad to do this comparison and never mention it at all in the manuscript.

We thank the reviewers for their speedy response and constructive feedback; please find below our responses to the comments.

Reviewer #1 (Remarks to the Author):

Major Comments:

1. Can authors quantitatively comment on the underlying factors that lead to the performance gain in scPair by setting up some head-to-head comparisons? Is that because scPair uses all features without pre-selection before inputting the model, or is that due to the cross-modality structure of the scPair model? The former should be easy to quantify by training two scPair models with and without feature selection before feeding and comparing the performance difference. This will be a critical insight in the light of shaping a more robust preprocessing workflow for the overall single-cell analysis, especially for scATAC. The latter question about the model structure advantage can be achieved by comparing scPair to a more similar model, for example, scMDC (PMID: 36513636) that was also published in Nat Comm. scMDC's model structure also includes the ability to project one modality to the other and has an encoder layer that implicitly selects the most important features for the two modalities, making it more comparable to scPair. On the other hand, scMDC concatenates the two modalities before the encoder layer so that the encoder layer is a mixture. This key difference will help us understand the benefit of training two FNNs and a connecting NN (e.g. scPair) over training a single GAN (e.g. scMDC).

As suggested, we have conducted further benchmarking by training two sets of scPair models across various datasets from different species and tissues: one set without pre-feature selection and the other with pre-selection based on highly variable features. The results, now included in the revised manuscript (Fig S3), demonstrate that performing feature selection before mapping leads to worse mapping between the learned representations/cell states of both modalities (RNA and ATAC in this case).

Regarding how much of scPair performance is driven by model structure, our initial benchmarking included comparisons with state-of-the-art (SOTA) deep learning models such as multiVI and Cobolt, which also employ modality-specific FNNs (encoders) to project each cell's multimodal profiles into low-dimensional embeddings. scPair outperformed these models, highlighting the advantages of its architecture and optimization strategy.

Still, to address the specific suggestion, we have now further benchmarked scPair against scMDC, a model with similar functionality but which concatenates both modalities before the encoder layer. scPair outperforms scMDC on 6 out of 7 of the benchmark datasets in predicting one modality to the other (feature prediction), except in the case of the SNARE-seq2 human cortex sample (both methods achieved decent performance on this dataset) (see Review Table 1 below).

However, we note that scMDC has label leakage issues; for instance, when predicting the RNA profile, both RNA and ATAC inputs are required, which results in leakage of the target label (RNA). This makes scMDC unsuitable for cross-modal prediction tasks. In contrast, models like scPair, Cobolt, Polarbear, and MultiVI can predict one modality

from the other independently, making them applicable to unimodal data. Given this significant limitation, we believe that a direct comparison between scMDC and these other methods is not fair. Therefore, we have decided not to include these results in our manuscript, but we provide the benchmarking results here in response to the reviewers' comments.

	scMDC	scPair
sciCAR cell lines	0.447	0.675
10x Multiome human PBMCs	0.717	0.930
10x Multiome human Neurons	0.511	0.903
SHARE-seq mouse skin	0.858	0.891
SNARE-seq mouse cortex	0.698	0.723
SNARE-seq2 marmoset cortex	0.688	0.704
SNARE-seq2 human cortex	0.894	0.821

Review Table 1. The RNA prediction performance (average Pearson's correlation) of scMDC and scPair on the held-out cells across seven single-cell multiomics datasets.

2. It is unclear how the authors implemented “the dimensionality of the encoding hidden layers and modality-specific or joint representation layers of all benchmarked methods were set to match the cell state dimensions in scPair” (Line 562-564). Is it done at the model-setup stage before training or at the post-training stage, where the encoder space is truncated to fit the same size as that in scPair? In fact, different models' latent spaces may have learned different “biological” contexts. For example, the RNA encoder layer of scPair learns the latent features that are most representative of ATAC, but in Polarbear it learns the latent features that are most representative of gene expression. Therefore, forcing the same dimensionality as scPair won't make it fair for other methods. If the authors kept other tools' optimized dimensionality in the latent space and only re-compared the performance in all the matrices, would the conclusion change systematically (e.g. Figures 2c-d and 3a-b)? This should be at least included as a supplementary.

To clarify, the dimensionality of the latent space of each method was set to match the cell state dimensions in scPair at the model-setup stage, prior to training (in the initial submission).

We agree that different models' latent spaces may capture different biological contexts, and forcing the same dimensionality may not provide a fair comparison. Therefore, we have re-evaluated the performance of each tool using their reported optimal latent space dimensionalities and training procedures and use these results in the updated manuscript (Methods section, **Execution of generative model-based methods for single cell analysis**). The results demonstrate that our conclusions remain consistent with our previous results (Figures 2-3, Supplementary Figures S2 and S4), where scPair outperforms or achieves competitive results compared to the SOTA methods across all the benchmarking datasets.

3. I had a bit of a hard time getting my head around: both Fig. 2a and Fig. 3a compare the performance of RNA-to-ATAC projects across tools on different datasets. While I understand the two figures using two different matrices, namely (1-FOSCTTM) and Pearson correlation, why the overall trends are quite consistent on the four SNARE-seq datasets but different on 10X? Especially, why Polarbear is close to scPair on 10X in

Fig. 2a but suddenly far away from scPair on 10X in Fig. 3a? Theoretically, (1-FOSCTTM) and Pearson correlation should be correlated/translatable?

The reviewer is correct that in the initial submission, the relative performance results are highly consistent across the 3 SHARE/SNARE-seq datasets, yet differ when compared to the 10x dataset results in Figures 2 and 3. We believe this discrepancy is largely driven by systematic differences in sparsity, sample size, and heterogeneity between the platforms, particularly in the ATAC modality. We observed that the non-zero peak counts within the three SNARE-seq mammalian cortex datasets account for only 2-3% per matrix, while they account for over 6-8% in the 10x datasets (as shown in Supplementary Figure S4b); in other words, SHARE/SNARE-seq likely suffers from a higher dropout rate compared to 10x Genomics assay. We expect the sparsity of the measurements plays a significant role in dataset performance.

To further investigate this, we included an additional 10x dataset on neuronal samples, where the non-zero counts for ATAC modality are approximately double those of the SNARE datasets on average, as suggested by Reviewer 2. We also included another sciCAR dataset with much higher sparsity. To address this fairly, as per Reviewer 1's comment, we re-ran these methods with their optimal embedding settings and updated our benchmarking results. The updated results are shown in the Figures 2-3, and both scPair and the SOTA method MultiVI are robust across different datasets.

Regarding Polarbear's performance, we emphasize the two metrics reported in the original Figures 2 and 3 are measuring fundamentally different quantities. Namely, the 1-FOSCTTM metric in Figure 2 evaluates the accuracy of the latent space, while the Pearson correlation in Figure 3 evaluates the predictions of original data features (genes and peaks). In Figure 2, the 1-FOSCTTM is computed over an $N \times K$ matrix, where N is the number of cells in the held-out set, and K is the size of the latent space. In contrast, in Figure 3, the Pearson correlation is computed over a $N \times G$ feature matrix, where N is the number of cells and G is the number of genes, and $G \gg K$. The results in original Figures 2 and 3 are thus measured over very different matrices, explaining the potential discrepancies in some extent. We also re-ran Polarbear using its optimal settings on all datasets, and its RNA prediction performance is still relatively poor on the 10x PBMC dataset, and the results can be found in the SourceData file (under Fig2 tab, C10).

4. I'd encourage the authors to improve the text for Figure 3e-g so that the general audience who are not familiar with the benchmark datasets can easily recognize the power and robustness of scPair. Specifically, what are the tissues and expected cell types/clusters? What are the known gene markers of each type/cluster and how do the four ATAC regional examples selected by the authors support those expected gene markers? The authors' own description of neonatal mouse brain SNARE-seq dataset is a good exemplar to follow.

We have rephrased the figure captions and text. The tissue and cell type are indicated in the figure captions, and we now also point this out in the main text, together with the marker genes (Rasgrf2, Rorb, Sulf1, and Tie4) of the cell types and how we select the peaks for visualization in the panels f and g.

Revised text (can also be found in the updated manuscript in lines 201-213):

As an additional set of visualizations to evaluate the quality of predicted per-chromatin region accessibility profiles, Figure 3e illustrates a UMAP visualization of the predicted single cell chromatin accessibility cell state derived from RNA input of held-out test cells in the SNARE-seq mouse cortex dataset. These cells are labeled by the predefined cell type from the original study, and includes cell types such as cortical layer 2-3 and layer 5-6 excitatory neurons, among others. Globally, the predicted ATAC cell states successfully capture the separation of cell types in these held-out cells not used for training, suggesting that scPair effectively projects RNA to ATAC cell states. At the individual locus level, we identified four chromatin regions whose accessibility patterns were identified as markers of cell types (Fig. 3f); we see their predicted accessibility patterns in the held-out samples (Fig. 3g) are consistent with their corresponding cell types (Fig. 3e) (excitatory neuron layer 2-3 and layer 5-6 subtypes). These results support the robustness of scPair in predicting individual chromatin region accessibility patterns accurately, using only the paired scRNA-seq data.

5. In Fig. 4e, it is impressive to see the strong alignment of key gene expression between the scPair pseudotime and the real-world development time. However, I have some questions about the selection of these key genes: 1) first of all, I am wondering why not all genes are colored according to the development stages, which are supposed to be purple, red, green, blue and orange according to other Fig. 4 panels. What do the gene names in black mean? 2) the authors claimed these genes are adapted from Chen et al.'s Nat Biotech paper in 2019. But some genes in Fig. 4e are NOT the markers of the five stages in the original studies. For example, Vim was reported by the original authors as a marker for Ast/RG, not any of the five stages. And Notch1 was NOT in the original marker list. These discrepancies might be due to the fact that the author re-analyzed the original data by subsetting to five stages instead of using the whole data. If the authors used the markers of the five stages from the original study's Supplementary Table 1, which I think is a more robust estimation with more cell types involved, what would the new Fig. 4e look like compared to MultiVI's outcome? The reviewer is correct – in the original figure, we have used a mixture of genes from both the original paper (Chen et al., 2019) as well as some re-analysis papers. We have modified the Figure 4e (now Figure 4c) heatmaps to use the gene list provided by 2019 Chen et al.'s Nature Biotech paper only. In the main figure, the marker gene list is from the original papers' Supplementary Figure 14a, and the genes are ordered by the cell types. The heatmaps with the full marker gene list (based on Chen et al.'s Supplementary Table 1) are in Figure S6 now, as the reviewer suggested. Based on the updated results, our previous statement is still supported.

We agree the coloring of the names is misleading. Now, we are not coloring any gene names.

6. While the authors described a complex process of their approach of integrating unimodal scATAC data into their scPair models to improve the performance from Line 250-255. But the language there needs to be corrected and orchestrated in a better

way, as this is the most critical technical part that will help the readers to digest the rest of the paper. Currently, it reads not only repetitive/circulated but also broken.

For example,

1) “We first perform trajectory inference on the RNA component... more successful in RNA space” which is fine, but the next sentence, “The RNA cell states used as input to trajectory inference are...”, talks about the same thing again and seems to be unnecessary.

2) “Next, we predict the RNA cell ...” is fine. But the next immediate part, “use the trained scPair model to infer the cell state of each cell in the unimodal scATAC dataset” comes from nowhere.

Is this a part going to be removed? In addition, I’d also suggest the authors include a main figure here to facilitate the demonstration. (This can be Fig. S1d if taking Q13 below into account is feasible.)

We have rephrased this part as (can also be found in the updated manuscript in lines 263-272):

Here, we assume we have a (smaller) multimodal RNA and ATAC dataset, as well as a larger unimodal scATAC dataset. In both strategies, we initially train scPair on the multimodal data only. Then, in the first strategy, we subsequently pass the RNA and ATAC-based cell states estimated for the multimodal data only to an existing trajectory inference method⁶¹ to estimate pseudotimes of each modality separately, yielding the pseudotime estimates $scMultiome_{RNA}$ and $scMultiome_{ATAC}$. This strategy represents data analysis using only the smaller, multimodal dataset. The second integrated strategy leverages a larger unimodal scATAC-seq dataset by estimating ATAC cell states using the trained scPair model, and passing them through the trajectory inference framework⁶¹ to assign each unimodal scATAC-seq cell their $scUnimodal_{ATAC}$ pseudotime (Fig. S1d).

7. The number of cells involved in the scPair’s implementation on the Allaway et al. dataset is quite confusing and needs to be better clarified. Line 268 says there are 24,589 unimodal scATAC cells. But Line 280 says the pseudotime assignment is only to 14,605 unimodal scATAC cells. Is this because the missing (~10,000) unimodal scATAC cells have been consumed in the training process? It sounds contradictory to Line 258 where the authors claim that “each unimodal scATAC-seq cell will have an assigned pseudotime”.

We agree the description of the number of cells from the Allaway study is confusing. The discrepancy arises from the fact that in the original study, they generated single-cell profiles for both mitotic and post-mitotic cells, resulting in a total of 24,589 unimodal scATAC cells. In their analysis of developmental trajectories and cell fate decisions, they only focused on the 14,605 post-mitotic subset of cells, and we followed their strategy in our manuscript. To avoid confusion, we have rephrased the relevant sections to only use the 14,605 number (of post-mitotic cells).

8. What is the downsampling strategy used in “Pseudotime-binned pseudobulk profile” toward the bin with the lowest counts? Can the authors comment on whether the cells with important features might be filtered out during this process? In other words, can the

authors justify their downsampling strategy would lead to reproducible results in Fig. 5d?

Our downsampling strategy is designed to equalize the number of cells across all bins for the pseudobulk profiles. Since some bins start with more cells than others, to make them equal, we identified the bin with the fewest number of cells (N_{minimum}), and simply subset other bins to have the same number of cells (N_{minimum}) cells. For bins with more than N_{minimum} number of cells, we select N_{minimum} number of cells uniformly at random. We therefore would not expect bin-level features to be filtered out in the process since we select uniformly at random.

To further address the potential concerns about downsampling in Figure 5d,e (now Figure 5e,f in the updated manuscript), we have now instead averaged the accessibility across all cells within each bin (without downsampling), and the conclusions are still consistent with the original figure, suggesting our specific downsampling strategy was valid.

9. Fig. 5e bottom panel, does the x-axis actually represent the pseudo-time of “RNA cell states predicted from” Unimodal scATAC? If so, it should be labeled precisely. The current label of “Unimodal scATAC-seq pseudotime” is misleading and has no difference from that in the bottom panel of Fig. 6d. Similarly, in Fig. S7d, the x-axis seems better to be “Pseudo time based on RNA cell states predicted from 10x scMultiome ATAC”. For Figure 5e (now Figure 5f in the updated manuscript), the x-axis of the bottom heatmaps represents the pseudotime inferred from the ATAC cell state using the unimodal scATAC-seq data. Neither the measured unimodal RNA data nor the RNA cell state is used here. The x-axis represents the same pseudotimes as in the bottom heatmaps of Figure 5e. However, the values in the heatmap are based on RNA prediction from the unimodal scATAC-seq data. To avoid confusion, we have relabeled it as “Unimodal scATAC-seq pseudotime (predicted RNA profile)”. Similarly, for Figure S7d (now S8e), the legend indicates “predicted RNA expression” of bimodal data.

10. Line 309, the author writes “The improved identification of branch-specific genes is not driven by our use of scPair”. I might get where the authors are trying to reach with this sentence, but I don’t agree with the current way of expressing it.

In contrast, the improvement is indeed driven by the scPair by leveraging the unimodal data.

The failure to display the off-diagonal trends in the heatmaps in Fig. S7d might indicate that the training of scMultiome ATAC → RNA projection was not successful due to the issue of data quality/low cell number; yet scPair can help to overcome this caveat by borrowing information from unimodal scATAC. This is only my hypothesis, and the authors still justify whether it is correct.

We have reworded this part as (can also be found in the updated manuscript in lines 333-343):

Because our comparison was between two different uses of scPair, the improved identification of branch-specific genes is driven specifically by our integration of the larger unimodal scATAC dataset, rather than being driven more generally by the use of scPair to predict RNA states of the unimodal scATAC-seq data. To further demonstrate

this, we repeated our above analysis except we only used the ATAC component of the 10x scMultiome cells to predict RNA profiles and infer pseudotimes (instead of the larger unimodal scATAC-seq dataset). As shown in Figure S8e, the results of this 10x scMultiome-based analysis do not capture branch-specific gene expression trends as well as when we use the larger unimodal scATAC-seq dataset (Fig. 5f, bottom), suggesting our improved identification of branch-specific genes is driven primarily by the use of a larger unimodal scATAC-seq dataset.

11. Fig. 5g provides highly valuable insights into neurodevelopmental regulation. To enhance the excitement of scPair application and the biological impact of these findings, can the author also provide a list of genes associated with each set of chromatin openness? Specifically, if each set of chromatin openness is forced to be zero in the ATAC → RNA FNN, which set of genes is most influenced on the output side? We appreciate the reviewer's enthusiasm for the insights into neurodevelopmental regulation provided by Figure 5g (now Figure 5h). To clarify, the heatmap illustrated in Figure 5h represents motifs (rows) that were found enriched across open regions of specific cells (columns), where the columns are arranged based on estimated pseudotime. As suggested, we now include the row names corresponding to the enriched motifs in the SourceData spreadsheet.

We interpret the reviewer's comment to suggest identifying the target genes of the enriched motifs (since the motifs are identified by looking at open regions of a given cell). We cannot directly identify target genes from the enriched motifs in this analysis, as the enrichment computed by chromVAR is based on global enrichment across all open chromatin regions of a cell; chromVAR does not return a list of the individual open regions that lead to the detected enrichment, and it is not clear what is the implicit "threshold" of how many instances of each binding motif in a region is necessary for the corresponding TF to bind to that region. Strategies such as co-expression analysis can be used to hypothesize what the target genes of a given TF is, but we did not include such an analysis here because of the complexities of regulatory network inference and analysis, and the fact it is somewhat tangential to the central goal of the paper (scPair).

12. The gain from scPair-augment on Patch-seq is quite modest. This seems to suggest that even if the sequencing depth in the Patch-seq is low, they still robustly measure the key features that largely contribute to dimension reduction. While this is not to say scPair-augment is totally useless, the authors may want to retain Fig. 6c-d in the supplementary figures.

We acknowledge the reviewer's observation that the gain from scPair-augment on the Patch-seq data is modest compared to other datasets. We do want to note that the Patch-seq data actually has a high sequencing depth comparable to that of unimodal RNA atlases (as shown in Figure S7c right), but a relatively low number of samples. While the gain from scPair-augment may be modest in this specific case, we envision there may be other scenarios (perhaps outside of the single cell/nucleus sequencing field) in which augmentation can be successful, and think the readers would benefit from being aware of the possibility of using unimodal data in this way. We agree with

the reviewer's suggestion and have moved the original Figures 6c-d to the supplementary document (Supplementary Figure S9c-d).

13. A final thought and not necessarily to be addressed: given the inclusion of unimodal scATAC indeed leads to an improved biological interpretation on the Allaway et al. dataset, I am wondering whether the authors can

1) absorb scPair-augment implementation on the Allaway et al. dataset and replace the scPair-standard there, and

2) land on all the same conclusions as in the current Fig. 5.

If so, that can be an ideal place to establish the power of scPair-augment, against low cell number instead of against low sequencing depth, with critical and novel biological insights.

In Figure 6, we demonstrated the advantages of scPair-augment in addressing both lower cell numbers (SNARE-seq data and Patch-seq data) and lower sequencing depth (SNARE-seq data). Building on these findings, we agree with the reviewer that applying scPair-augment to the Allaway et al. dataset and recapitulating the critical biological insights from Figure 5 would provide further convincing demonstration of our method's capability. We tested the scPair-augment model and found the augment vs standard pseudotime consistency (correlation) to be 0.96 for the unimodal dataset. As a result, we believe there is not much difference, and the results should be consistent.

Minor Comments:

1. The author should provide better guidance on installing the scPair dependencies. Specifically, I found that three packages, namely "intel-openmp", "cobolt" and "jaxlib", couldn't be installed directly from the provided yml file. I had to install "cobolt" and "jaxlib" manually. I still couldn't install "intel-openmp" and am wondering if it is necessary.

Thank you for bringing this to our attention. We have re-generated the dependency requirements for running scPair, which can be accessed through (<https://github.com/quon-titative-biology/scPair/blob/main/scpair.yml>).

The "cobolt" package was only used for benchmarking experiments and can be installed from their GitHub page (<https://github.com/epurdom/cobolt>).

As for "intel-openmp", it is not a necessary dependency for running scPair, and we have removed it from the updated requirements.

2. The authors interchangeably wrote Polarbear, PairedVAE, and Polbear/pairedVAE in the text and figure legends. Please use a consistent nomenclature.

We have carefully reviewed the text and figure legends and have corrected the instances where the nomenclature was inconsistent. The method will now be consistently referred to as "Polarbear" throughout the manuscript.

3. Is there a reason Cobolt's performance is not included in Figure 3b?

While the other benchmarked methods used binarized chromatin accessibility for modeling, which allowed us to use AuROC to evaluate the performance of "missing

modality" (ATAC profile) prediction, the Cobolt method employed a latent model for chromatin accessibility inspired by the Latent Dirichlet Allocation (LDA). As Cobolt does not output ATAC predictions in the form of binarized profiles, its performance is not directly comparable to the other methods in Figure 3. We have clarified this in the results section.

4. The authors repeatedly cite the same methodology and data source papers, which I think is unnecessary. To name a few, check the occurrence for references 33, 59, and 60. I'd suggest citing them once they are first mentioned in the main text. Then keep using the same term throughout the paper without citing them again.

We have reviewed the manuscript and removed the redundant citations for the same methodology and data source papers.

5. Figure S6, the author names are used on the plot but the database names are used in the legend. It is better to unify them for easy reading and understanding.

We have unified the plot labels in Figure S6 (now S7) to match the figure captions by removing the author names, ensuring consistency between the legends and the plots for better readability and understanding.

6. Line 282, "three terminal terminals"?

We have corrected it by removing the redundant word.

7. Fig. S7b and S7e-f are not mentioned in the main text. I believe Fig. S7b should have been cited along with Fig. 5b, c, and Fig. S7c in Line 289.

We have reviewed the main text and included the appropriate citations for the plots mentioned in the relevant sections.

8. Throughout the section "Unimodal datasets increase the density of cells along multimodal trajectories", the word "scATAC-seq" sometimes is used alone without the attributive "unimodal" in front, which might cause confusion.

We have reviewed the section and ensured that "unimodal" is consistently used before "scATAC-seq" when referring to the unimodal data.

9. Fig. 6b is better plotted according to (1-FOSCTTM) to visually display gains from using unimodal data. This way is also consistent with Fig. 2c.

We have updated Fig. 6b to plot (1-FOSCTTM) on the y-axis.

10. Due to the amount of datasets used in this paper, can authors provide a supplementary table that associates the analyses/benchmarks with their involved datasets?

We have generated a SourceData file with a datasets spreadsheet that includes brief descriptions/statistics of the datasets, such as the number of cells, the number of features per modality, and the non-zero counts of each modality to reflect the sparsity of the input data. This supplementary table will help readers easily associate the analyses/benchmarks with the respective datasets involved.

Reviewer #1 (Remarks on code availability):

1. This refers to Q1 in my Minor Comments: Three dependency packages, namely “intel-openmp”, “cobolt” and “jaxlib”, couldn’t be installed directly from the provided yml file. I had to install “cobolt” and “jaxlib” manually. I still couldn’t install “intel-openmp” and am wondering if it is necessary.

We have re-generated a clean version of the environment YML file, ensuring that all necessary dependencies are included. Additionally, we have implemented easy-to-run functions that allow users to run scPair directly on scanpy objects with just a few lines of code, improving accessibility and usability.

2. The authors' codes and scPair's dependencies are based on Python 3.8. I would also encourage the authors to update them to Python 3.10, as many computing environments are progressively abandoning Python 3.8.

In the current version of our source code and environment, we have followed the recommendation and switched to Python 3.10.

Reviewer #2 (Remarks to the Author):

Major Comments

1. The authors state that “we intuitively expect that there should exist a mapping between RNA-based cell state and ATAC-based cell state”. According to my opinion chromatin changes occur before induction of transcriptional changes in time. Is the ATAC-based cell state than not a projection of the current and future RNA-based cell state? Will scPAIR takes this into account and if not, does scPAIR is not overcorrecting? The reviewer is correct that at the level of a single gene, chromatin changes of specific regions (e.g. regions containing regulatory elements governing transcription of the gene of interest) would be expected to occur before induction of transcriptional changes in time, thereby suggesting one could view ATAC measurements (of those specific regions) as informative of the future state of RNA. However, at the genome-scale, we would not expect the RNA state to simply be a lagging copy of the ATAC state. There are a number of reasons for this:

- In general, we don't know which regions (and their associated accessibility) contain the relevant regulatory elements for a given gene, and there are many regions whose accessibility are not correlated with known genes' expression levels. While there has been a lot of work developing computational methods to link non-coding regions to genes based on chromatin accessibility and other data types, this is a hard problem (Avsec et al., 2021, PMID: 34608324; Brennan et al., 2023, PMID: 37557175).
- Most single cell sequencing technologies capture a single snapshot of the modality they are sequencing (RNA, ATAC, etc) due to the need to destroy the cell during sequencing. Transcriptional regulation at different loci might be expected to have different dynamics, time-scales and conditions (or stimuli) that lead to regulatory elements opening and closing, TF recruitment and ultimately gene expression occurring. Therefore, for a single sequenced cell (and the corresponding RNA and ATAC cell state spaces we infer from them), it is not trivial to leverage the idea that ATAC changes might precede RNA changes at the genome-wide level.
- Our work (e.g. Figure S5c), as well as the work of others (Chen et al., 2019, PMID: 31015418; Xiao et al., 2024, PMID: 38493343), has shown that cell states are much more readily identified and separated in RNA space compared to ATAC space. While this may partly be a result of the fact that RNA measurements are a more direct and quantitative measurement of gene expression compared to ATAC (which is binary – a region can only either be open or closed), this observation still supports the notion that RNA and ATAC are not readily mapped to one another (even if RNA changes conceptually follow ATAC changes).
- Finally, our statement of expectation that there should inherently be a mapping from ATAC to RNA and back merely arises from the fact that for multiomic measurements, the RNA and ATAC data are being measured on the same cell --- since both measurements are from the same cell, then conceptually they should 'map' onto one another.

2. In the introduction there something mentioned about the higher depth of unimodal

data. This is chosen by the data generator and is not intrinsic to the methods themselves as you can define yourself how intensive you will sequence every pool. We have clarified in the text by using the term “effective depth/coverage”, rather than sequencing depth. The reviewer is correct in that in the experimental design of a single cell study, the experimenter controls the sequencing depth (both in terms of number of cells sequenced, and the average number of reads sequenced per cell). What we mean by “effective depth” is that, while the experimenter can control the average number of reads per cell, there is less control over (1) the actual number of reads per individual cell that end up being sequenced, and (2) in the case of e.g. 10x technologies, the number of UMI (unique transcripts) sequenced per cell, which is what we term ‘effective depth’ (as the number of UMIs per cell is what directly affects library size and sparsity of the measurements of the cell). It is well established that the “effective sequencing depth” can vary widely across cell types even within the same study, as there are both technical and biological factors that ultimately affect e.g. the PCR duplication rate for a given cell. Also, even though the experimenter has the control to increase sequencing depth per cell, there is a fundamental tradeoff between sequencing cells more deeply, and sequencing a higher total number of cells – for some applications (such as rare cell type detection), experimenters may choose to sacrifice sequencing depth per cell for larger numbers of cells. What we have observed in our data (Supplementary Figure S7) is that for various reasons, multimodal (ATAC + RNA) single cell datasets tend to have lower effective depth than unimodal datasets, which is partly what motivated the development of scPair.

3. For most datasets there is a big difference in the mapping accuracy and the prediction accuracy between the PBMC dataset and others. What is the reason for that? Is this intrinsic to the method? 10Xgenomics vs all others? Or is either sequencing depth related? A second 10X genomics dataset would be of value to see if it is related to the methodology or the sample type.

This is likely primarily due to the fact the 10x Genomics human PBMCs data is of good quality (higher sequencing depth, lower sparsity/technical noise) compared to the SNARE-seq datasets. To address the reviewer’s concern, we have added a second 10x Genomics dataset into the benchmarking and provided SourceData tables to describe each benchmarked dataset including number of cells, number of features, and non-zero counts (reflecting the sparsity). Also, we used the optimal settings for each method and re-ran all the experiments; our conclusions about the superiority of scPair still hold with the new results and datasets.

4. How scalable is scPAIR (both modes) to large datasets (100k->1Mcells)?

Currently, in the literature, there are no single studies with paired scMultiome (ATAC + RNA) data reaching 100k or 1M cells to our knowledge; our largest benchmarked dataset, the SNARE-seq2 human cortex data, only contains approximately 85k cells. To address the reviewer’s concerns, we compared the run times of scPair to other methods (using their default settings) on our benchmark datasets (Figure S4a, also reproduced below). Note that scPair (purple line) is the fastest method on the largest datasets. The longest training time for scPair was 23 minutes on the ~85k cell dataset with over 50k

features in total. The corresponding table can also be found in the SourceData spreadsheet.

Moreover, we tested a bridge integration method the reviewer recommended, StabMap (PMID: 37231260), known for its fast speed. For example, when tested on the sciCAR (4k cells) dataset, it took less than 20 seconds. However, in addition to the cell number, the feature number also influences model speed and scalability. Notably, scPair does not apply any feature pre-selection steps, whereas StabMap does according to the tutorial. When using the full gene and peak features as scPair does, StabMap took 9 hours to complete the run on the sciCAR (4k cells) dataset. Since the benchmarked methods in this manuscript are deep learning-based and benefit from GPU acceleration, we did not include StabMap in the run time comparison to ensure fairness.

a

b

	number of cells	number of RNA features	number of ATAC features	RNA non-zero counts pct.	ATAC non-zero counts pct.
sciCAR cell lines	4,216	17,368	57,425	8.43%	0.55%
SNARE-seq mouse cortex	8,055	9,104	37,030	10.17%	3.52%
10X Multiome mouse brain	9,370	14,461	82,474	27.33%	5.76%
SNARE-seq2 marmoset cortex	9,946	20,292	35,573	18.85%	2.25%
10X Multiome human PBMCs	11,331	13,515	86,002	14.60%	8.09%
SHARE-seq mouse skin	34,774	15,436	97,669	4.15%	3.15%
SNARE-seq2 human cortex	84,178	16,139	33,517	13.67%	2.03%

5. The number of cells and reads per cell clearly have an impact on the integration (see trajectory part) thus what is the minimum number of cells (for the “bridging population”), read coverage needed to perform good enough?

The reviewer is correct in that the number of cells and coverage per cell are crucial factors that impact the quality of integration in trajectory analysis. Intuitively, in single-cell trajectory studies, it is essential to ensure that the trajectory is adequately covered by a sufficient number of bridging cells to capture the dynamic processes accurately. However, the exact minimum number of cells and read coverage needed can vary depending on several factors, such as the complexity of the underlying biological processes, the diversity of cell states, the kinetics of the continuous process (e.g. differentiation) and the evenness of cell sampling along the trajectory. The question of the precise number of cells needed therefore depends on the system under study in short, similarly between multimodal versus unimodal studies. The only statement we are comfortable making is that if the assay used to generate the multimodal dataset (e.g. 10x Multiome) is producing lower effective depth cells compared to their unimodal counterparts, then the number of cells needed for a multimodal dataset would be expected to be at least as large (if not larger) than that needed for the equivalent unimodal dataset.

The question of “how many cells does the bridging population need” is similar to the question of how many cells does one need to capture all the relevant cell types in a particular sample; it is dependent on the particular sample one is sequencing, and in the experimental design, one has to use a combination of prior knowledge, pilot sequencing studies on a few samples, and existing literature to figure out how many cells are needed.

6. scPair augment is not benchmarked against methods that work with a bridge population. Is this strategy better performing compared to similar methods? To address the reviewer’s concern, we have now included one bridge integration method StabMap (PMID: 37231260) in our paper. Note there are systematic differences between scPair and StabMap (scPair is deep learning-based that leverages the full input feature set, while StabMap is a fast method with similar functionality but uses log-normalized profiles with a feature selection step). We therefore compared the standard scPair (not augmented) with StabMap on the SNARE-seq mouse datasets (ATAC+RNA) using both methods’ default settings (details can be found in Methods section) and also benchmarked both methods with and without incorporate large unimodal datasets in Figure 6b.

7. In the discussion the authors suggest that the method might work on other multimodal datatypes Eg. CITE-seq data, nanoCUT&TAG data, ... I think this should be proven or otherwise it should be formulated less strong and possible struggles to use it for other methods should be mentioned more in the discussion (eg, adaptation of distributions, ...).

We have applied scPair in Patch-seq data in our manuscript (Figure 6c and Supplementary Figure S9b-d), where the electrophysiological modality has a different distribution from RNA (NB/ZINB) and ATAC (Bernoulli).

To further address the reviewer’s concerns, we conducted additional experiments using scPair on CITE-seq data and presented the results in Supplementary Figure S10 (also

shown below), demonstrating scPair’s potential application to other types of single-cell multimodal datasets (RNA + ADT). We achieved modality-specific cell state spaces with satisfactory cross-modality mapping accuracy (a). The clustering separation based on the cell states learned by scPair is biologically meaningful, as indicated by the high expression of the CD14 marker gene and surface protein in the same distinct population, which is separate from other cells with little to no CD14 expression (b).

8. The documentation on how to perform scPAIR is not complete yet. The example code should be better documented. Ideally scPAIR starts from a multimodal anndata object and is performed in only some lines of code (e.g., only splitting data, training, testing). Now it looks like the user has to still code quite a lot.

We have updated the tutorial code for running our method. The demo input data is also provided in our GitHub repository with the link to figshare (https://github.com/quon-titative-biology/scPair/tree/main/sample_data, or <https://figshare.com/s/ea98335f6f8a0abc8ae5>). The data is saved as scanpy/AnnData objects in h5ad files, which are easy to load and run using our updated function code. Preparing the data splits and running scPair can be accomplished with a single function each.

Minor Comments

1. Figure 1c: towards instead of towards
Typo corrected.

2. Figure 3: put y-axis from 0-1 on both A and B to make this comparable
For the original Figure 3b, the auROC baseline is 0.5 (random guess), so we used 0.5-1 rather than 0-1. Now, we applied alternative visualizations to make sure the updated Figure 3a and 3b more comparable.

3. From the introduction and abstract it should be more clear which type of multimodal

data is needed. By reading the text both paired and also unpaired with bridge populations is possible.

We have updated the introduction and abstract to clearly indicate the paired bridge multimodal data is needed. Here is the updated abstract:

Single cell multimodal assays profile multiple sets of features in the same cells and are widely used for tasks such as identifying cell type and cell state, mapping chromatin state to RNA state and vice versa, and linking non-coding regulatory elements to their target genes. The analysis of multimodal single cell data is challenging due to two factors: high dimensionality of the input features relative to the number of cells sequenced, and shallow sequencing depth compared to unimodal assays, both of which hinder data analysis.

Here we present scPair, a framework for multimodal single cell analysis that leverages an implicit feature selection approach to address these challenges. scPair consists of a pair of encoder-decoder structures that individually compute cell state embeddings based on one data modality, and is trained using paired single cell multimodal data. The principal goals of scPair are to align cell states across modalities, and predict features of one data modality from the other. We demonstrate that scPair achieve superior performance and faster execution time compared to existing methods on these tasks, as well as facilitate other downstream tasks such as simultaneous trajectory inference across data modalities. Furthermore, we demonstrate how scPair can augment smaller multimodal datasets with larger unimodal datasets to increase statistical power, and show with this power we are able to identify novel transcription factors that activate during different stages of neural differentiation. More specifically, we identify different groups of transcription factors that may function to initiate the differentiation program, consolidate neuronal identity, and prime cells for branch-specific maturation, respectively. scPair therefore provides a generalizable approach to overcome key challenges in emerging multimodal single cell research.

4. The introduction part is not always clear/to the point.

Line 42: “and most other common tasks” is vague,

We have deleted the redundant wording “and most other common tasks” as we mentioned the tasks such as data visualization, clustering, cell type identification, and trajectory inference already prior to it.

5. Line 53-55: about clustering unimodal data and defining parameters, this is also the case for multimodal data and is not changed after integration.

The point we intended to make is that by integrating multimodal data, we can obtain a more holistic view of cell subpopulations considering multiple layers of biological information.

6. Line 68-69: Sentence is not clear to me. You nowadays have also diagonal integration methods. Some of them use indeed bridge data, but others don't.

We now mention and add citation about the diagonal integration methods into this part.

7. Line 173-177, instead of mentioning the increase in Pearson correlation and auROC,

it would be better to mention the (average) values. It's a difference when it goes low to mid range value of mid range to high, which you can not conclude now from the text. We now utilized the ranking and dot plots in Figure 3 to provide more direct visualization. Also, the values (as well as the average values) are provided in our SourceData file.

8. In the method section I would like to see additional information for the used datasets for benchmarking. How many reads (RNA/ATAC) per cell for each modality, some QC parameters (median/mean mtRNA%, TSS overlapping reads, ...) for both modalities, .. We now include the QC parameters in the Methods section. For datasets without cell filtration QC steps, these datasets are processed once downloaded. Detailed descriptions of the benchmarked datasets can be found in the SourceData file.

Reviewer #2 (Remarks on code availability):

The documentation on how to perform scPAIR is not complete yet. The example code should be better documented. Ideally scPAIR starts from a multiomics anndata object and is performed in only some lines of code (eg, only splitting data, training, testing). Now it looks like the user has to still code quite a lot.

I did not try to run the code myself at the moment. There is no real example dataset / analysis provided yet.

We have integrated scPair with scanpy to allow users to run the analysis directly from a multiomics AnnData object with just a few lines of code. Additionally, we now provide well-documented example datasets and tutorials in our GitHub repository (<https://github.com/quon-titative-biology/scPair/>) to improve the usability and accessibility of our method.

Reviewer #3 (Remarks to the Author):

Questions:

1. p 6, line 142: why are you using the FOSTTM score instead of the usual Sensitivity/Specificity or ROC or Accuracy/PPV? I may be wrong, but I thought this was a simple class prediction problem, where the classifier predicts a label.

The reviewer is correct that in classification problems, one can use the beforementioned metrics. Here, we want to avoid explicit assignment of cells to predefined cell types. The fraction of samples closer than the true match (FOSCTTM) is widely used for evaluating the mapping accuracy when there is no label for clusters or cell types (originally from PMID: 33954299, and is also used for benchmarking paper PMID: 38493343). In this case, we are not assuming any cell type/class labels, just calculating the pairwise Euclidian distance between the ground truth points and the mapped/projected points.

2. p 12 | 323: You write that the motif NKX2-1 is specific for a certain type of non-coding regions. the motif of NKX2-1 is CACTTA (see UniProbe), but as far as I know, most NK-class homeodomain recognize this motif and similar ones. I am surprised that you can derive a hypothesis like this from just motif matches. Could you check is NKX2-1 is really the only predicted binding factor? I imagine there are a lot of other motifs that come up and you picked this one?

The reviewer is correct that there in general can be multiple TFs with highly similar motifs, making the mapping of motifs to factors difficult. Nkx2-1 is not the only predicted binding factor here, the reason why we highlight it because it was mentioned (with an alternative hypothesis) in the original paper. In our scPair-based re-analysis of the data, we noticed that this motif in not only enriched in the branch terminal stage (as suggested in the original paper) but also in an earlier stage. We have more motifs for that branch stage, and we have listed the names in the SourceData spreadsheet corresponding to Fig5.

Discussion:

1. I'm missing an explanation or hypothesis why ATAC seq data is mapping poorly to cell states. You have a hypothesis, could you add to the discussion?

We have added the hypothesis to the discussion regarding the poor mapping of ATAC-seq data to cell states. There are a number of possible explanations, including (1) the inherent sparsity and noisier nature of ATAC-seq data compared to RNA data, (2) the fact RNA is a more direct measurement of molecular functional activity of the cell compared to chromatin region accessibility.

Code and data sharing:

1. code needs documentation on how to load the input data and at least one example input file and output file.

We have updated our source code and tutorial to provide clear instructions on how to load the input data. Additionally, we have included a demo input dataset in the form of scanpy/AnnData objects saved as h5ad files, which can be easily loaded and run using our updated function code.

2. I think you can increase citations quite a bit if you can provide the dataset. Anyone

coming after you and who writes similar algorithms will need to do the same evaluation again. If you can provide the input matrices and the output prediction values for your and the other software packages will be able to use your files instead of spending a lot of time (x weeks?) on finding the input data from the other papers. You could upload them to figshare as a zipfile.

We appreciate the suggestion to enhance the reproducibility and reusability of our work. We have prepared the dataset files, which can be accessed via the provided link: https://github.com/quon-titative-biology/scPair/tree/main/sample_data_or <https://figshare.com/s/ea98335f6f8a0abc8ae5>. Additionally, the output files will be saved when running the code, following the instructions in the updated tutorial.

3. For example, I tried to find your input data on Nemo, you helpfully provide the link <https://assets.nemoarchive.org/dat-ek5dbmu> but then I don't know which of the hundreds of files you used for your analysis. If you had a figshare link, I could just download that.

Thank you for the feedback. To facilitate easier access to the processed data used in our analysis, we have prepared the h5ad files containing the relevant processed data (https://github.com/quon-titative-biology/scPair/tree/main/sample_data_or <https://figshare.com/s/ea98335f6f8a0abc8ae5>). The raw data for the Human and Marmoset SNARE-seq datasets can be found at the provided link (https://data.nemoarchive.org/biccn/grant/u01_zhangk/zhang/multimodal/sncell/), as mentioned in the data availability section.

typos:

- page 3, line 55: explicit -> explicitly
- page 5, line 09: it is derived based - not sure what is wrong here. Missing word?
- p.9 l254: missing word before the dot. not sure what it is. "We" ?
- p10 l274: we aimed to avoid -> we aimed at avoiding or rather: our aim here was avoiding
- p10 l284: we also illustrate *that*

Thank you for pointing out these typos above; we have corrected them.

Reviewer #3 (Remarks on code availability):

1. The conda command is great. However, in my environment, it took forever, as often these days with large conda packages. Not the authors' fault. A docker container would be very helpful.

We understand the challenges posed by large conda packages and their installation times. For the time being, we have focused on re-generating a clean version of the environment YML file.

2. The reproducibility section of the code is empty. The only file is fig1.py and it is empty. The second tutorial is missing. The main tutorial is nice but missing the input data. I don't know how the input data must be formatted and there is no description how to load the input data.

Thank you for highlighting these issues. We have updated our GitHub page by providing the necessary code and adding a full tutorial. We also included instructions on how to

format and load the input data, along with providing demo input datasets.

3. As for the data available, I've commented on the paper itself in the other input box. Please see the answers above in the Code and data sharing section.

Reviewer #1 (Remarks to the Author)

The authors have addressed all my comments nicely. I believe the paper is ready to be accepted by Nature Communications.

Reviewer #2 (Remarks to the Author)

The quality of the paper improved greatly. I am pleased that the authors added another type of multimodal data and included more multimodal datasets in the benchmark. The documentation on their GitHub is more extensive, follow-up with user comments will be key in the future. I only have a minor comment remaining.

- Fig 3A: For the score and ranking, it would be helpful to have some reference numbers and best/worse indication.

Thank you for your suggestion. We have now added symbols in Figures 3A and 3B to indicate the best performing methods on each dataset. Additionally, reference numbers have been included in the source data tables for further reference.

Reviewer #3 (Remarks to the Author)

The changes look good. Thanks for your detailed replies!

One note: Reviewer 1 had the great idea of asking for a comparison with scMDC, which you did and you found that the method is too different to be compared. However, could you still mention this fact and/or attach the results as a supplemental file? I guess that many readers will think that scMDC is very similar and if they see that you already covered this case, this would help them. It would be too bad to do this comparison and never mention it at all in the manuscript.

We added the results about to scMDC after "...note we did not include Cobolt in these comparisons because it does not output ATAC predictions as binary profiles as the other methods do, making it challenging to compare Cobolt to other methods.": We have further benchmarked scPair against scMDC, which concatenates features from both modalities before encoding them into the cell state space. scPair outperforms scMDC on 6 out of 7 benchmark datasets (Source Data). However, scMDC suffers from label leakage, as it requires both RNA and ATAC inputs for predicting RNA, making it unsuitable for cross-modal prediction tasks. In contrast, scPair, Cobolt, Polarbear, and MultiVI can independently predict one modality from the other.